



# Field characterization of the PM2.5 Aerosol Chemical Speciation Monitor: insights into the composition, sources and processes of fine particles in Eastern China

Yunjiang Zhang[1–4], Lili Tang[1,2], Philip L. Croteau[5], Olivier Favez[3], Yele Sun[6], Manjula R. Canagaratna[5], Zhuang Wang[1], Florian Couvidat[3], Alexandre Albinet[3], Hongliang Zhang[7], Jean Sciare[8], André S. H. Prévôt[9], John T. Jayne[5], Douglas R. Worsnop[5]

[1]Jiangsu Collaborative Innovation Center of Atmospheric Environment and Equipment
Technology, Nanjing University of Information Science and Technology, Nanjing 210044, China
[2]Jiangsu Environmental Monitoring Center, Nanjing 210036, China
[3]Institut National de l'Environnement Industriel et des Risques, Verneuil en Halatte, 60550, France
[4]Laboratoire des Sciences du Climat et de l'Environnement, CNRS-CEA-UVSQ, Université Paris-Saclay, Gif sur Yvette, 91191, France
[5]Aerodyne Research, Inc., Billerica, Massachusetts 01821, United States
[6]State Key Laboratory of Atmospheric Boundary Layer Physics and Atmospheric Chemistry, Institute of Atmospheric Physics, Chinese Academy of Sciences, Beijing 100029, China
[7]Nanjing Handa Environmental Science and Technology Limited, Nanjing 211102, China
[8]The Cyprus Institute, Environment Energy and Water Research Center, Nicosia, Cyprus
[9]Laboratory of Atmospheric Chemistry, Paul Scherrer Institute, Villigen PSI 5232, Switzerland

Correspondence to: Lili Tang (lily3258@163.com) and Yele Sun (sunyele@mail.iap.ac.cn)



## Abstract

A PM$_{2.5}$-capable aerosol chemical speciation monitor (ACSM) was deployed in urban Nanjing, China for the first time to measure in-situ non-refractory fine particle (NR-PM$_{2.5}$) composition from October 20 to November 19, 2015 along with parallel measurements of submicron aerosol (PM$_1$) species by a standard ACSM. Our results show that the NR-PM$_{2.5}$ species (organics, sulfate, nitrate, and ammonium) measured by the PM$_{2.5}$-ACSM are highly correlated ($r^2 > 0.9$) with those

measured by a Sunset Lab OC/EC Analyzer and a Monitor for AeRosols and GAses (MARGA). The comparisons between the two ACSMs illustrated similar temporal variations in all NR species between PM$_1$ and PM$_{2.5}$, yet substantial mass fractions of aerosol species were observed in the size range of 1–2.5 μm. On average, NR-PM$_{1–2.5}$ contributed 53 % of the total NR-PM$_{2.5}$ with sulfate and secondary organic aerosols (SOA) being the two largest contributors (26 % and

27 %, respectively). Rapid formation and thereafter growth of secondary inorganic aerosols (SIA) were observed under fog processing in NH$_3$-rich environments. Positive matrix factorization of organic aerosol showed similar temporal variations in both primary and secondary OA between PM$_1$ and PM$_{2.5}$ although the mass spectra were slightly different due to more thermal decomposition on the capture vaporizer of PM$_{2.5}$-ACSM. We observed an enhancement of SOA

under high relative humidity conditions, which is associated with simultaneous increases in particle surface area, gas-phase species (NO$_2$, SO$_2$, and NH$_3$) concentrations and aerosol water content driven by anthropogenic SIA. These results likely indicate an enhanced reactive uptake of SOA precursors upon aqueous particles. Therefore, reducing anthropogenic NO$_x$, SO$_2$, and NH$_3$ emissions might not only reduce SIA but also SOA burden during haze episodes in China.



# 1 Introduction

Atmospheric fine particles ($PM_{2.5}$, aerodynamic diameter $\leq 2.5$ μm) are of a great concern because they degrade air quality (Zhang et al., 2015a), reduce visibility (Watson, 2002) and negatively affect human health (Pope and Dockery, 2006). $PM_{2.5}$ also has potential impacts upon global climate change and ecosystems. However, the impacts remain highly uncertain, mainly due to their complex chemical and microphysical properties, and sources (Huang et al., 2014; Sun et al., 2014), and the unclear interactions between meteorology and atmospheric aerosols (Sun et al., 2015; Ding et al., 2016). Therefore, continuous measurements of aerosol particle composition particularly in a complete level with high time-resolution are essential to understand the variations and formation mechanisms of $PM_{2.5}$ and are important to validate and improve chemical transport models.

The aerodyne Aerosol Mass Spectrometer (AMS) (Jayne et al., 2000) is a state-of-the-art instrument for measuring size-resolved chemical composition of non-refractory submicron aerosol ($NR\text{-}PM_1$) with a high time resolution from seconds to minutes (Jimenez et al., 2003; Allan et al., 2004; Canagaratna et al., 2007). Organic aerosol (OA) measured by the AMS can be further deconvolved into various organic classes from different sources and processes using positive matrix factorization (PMF) (Paatero and Tapper, 1994; Lanz et al., 2010; Ulbrich et al., 2009; Zhang et al., 2011), which has greatly improved our understanding of the key atmospheric processes of OA during the last ten years (Zhang et al., 2007; Jimenez et al., 2009). Based on the AMS system, a simpler instrument, the Aerosol Chemical Speciation Monitor (ACSM), was designed and developed for robust long-term monitoring of the $NR\text{-}PM_1$ chemical species (Ng et al., 2011b; Sun et al., 2015). In China, the AMS and ACSM deployments for highly time-resolved chemical evolution processes of $NR\text{-}PM_1$ species in urban and rural areas grow rapidly since 2006 (Sun et al., 2010; Huang et al., 2010; Sun et al., 2012a; Xu et al., 2014; Zhang et al., 2015c; Sun et al., 2016; Wang et al., 2016b). The new findings and conclusions have been well



summarized in a recent review paper (Li et al., 2017). Secondary organic aerosols (SOA) and secondary inorganic aerosols (SIA = sulfate + nitrate + ammonium) have been found to be of similar importance in leading to the rapid formation and accumulation of $PM_{2.5}$ during the severe haze events in China (Huang et al., 2014; Sun et al., 2014; Zhang et al., 2014). Recent studies

have shown that heterogeneous reactions associated with high anthropogenic $NO_x$ and relative humidity (RH) levels are one of the major formation mechanisms of secondary aerosols, e.g., sulfate (He et al., 2014; Xie et al., 2015; Cheng et al., 2016; Chu et al., 2016; Wang et al., 2016a; Xue et al., 2016). One reason might be the aqueous oxidation of $SO_2$ by $NO_2$ in aerosol water is facilitated by the rich $NH_3$ which can partly explain the rapid formation of sulfate during severe

haze events in China (Wang et al., 2016a). Although the formation mechanisms of sulfate are relatively well understood, the impacts of aerosol water on SOA formation remains unclear (Xu et al., 2017b).

Limited by the aerodynamic lens, previous AMS and ACSM only measure aerosol species in $PM_1$. This is reasonable for the studies in the US and Europe where $PM_1$ accounts for a large

fraction (typically > 70 %) of $PM_{2.5}$ (Sun et al., 2011; Budisulistiorini et al., 2014; Petit et al., 2015). However, a substantial fraction of aerosol particles in 1–2.5 μm ($PM_{1-2.5}$) is frequently observed in China, and the contribution can be more than 50 % during severe haze events (Wang et al., 2015b; Zhang et al., 2015b). The source apportionment results of $PM_1$ might have differences from $PM_{2.5}$ by missing such a large fraction of aerosol particles. Therefore, the

instruments which can measure $PM_{2.5}$ composition in real-time are urgently needed in China for a better understanding of variations, sources, and formation mechanisms of $PM_{2.5}$. The techniques for real-time measurements of inorganic species have been well developed, e.g., particle-into-liquid sampler – ion chromatograph (PILS-IC) (Orsini et al., 2003), Monitor for AeRosols and GAses (MARGA) (Du et al., 2011), and Gas and Aerosol Collector – Ion

Chromatography (GAC-IC) (Dong et al., 2012), and also widely used for chemical characterization of $PM_{2.5}$ in China. However, real-time measurements of the total organics in



PM$_{2.5}$ and subsequent OA source apportionment were barely performed in China (Elser et al.,

2016). Although ambient organic carbon (OC) and elemental carbon (EC) can be measured

semi-continuously, typically in hourly resolution, they can only be used to differentiate primary

and secondary OC using EC-tracer technique (Turpin and Huntzicker, 1995). In addition,

size-segregated filter samples can provide a detailed chemical information in different size ranges,

but are greatly limited by the sampling duration, typically days and even weeks (Huang et al.,

2014; Xu et al., 2015; Ye et al., 2017). Therefore, real-time characterization of PM$_{2.5}$ is important

to have a better understanding of aerosol chemistry and sources of fine particles in

highly-polluted environments in China.

Very recently, a PM$_{2.5}$ lens that can transmit particles larger than 1 μm to the AMS and

ACSM detectors, has been developed and the performance has been fully evaluated in both

laboratories and field studies (Hu et al., 2016; Hu et al., 2017; Xu et al., 2017a). The results

showed that the PM$_{2.5}$-ACSM equipped with the new developed capture vaporizer (CV) can

detect approximately 90 % of the PM$_{2.5}$ particles, but more thermal decomposition of both

inorganic and organic species was also observed. Although the CV produces different

fragmentation patterns of organic and inorganic compounds compared with those of SV, it

reduces the particle bouncing effect at the vaporizer and hence improves the quantitative

uncertainties caused by collection efficiency (CE). The recent evaluation of the CV for inorganic

species measurements showed overall agreements with those by co-located measurements (Hu et

al., 2017). The PM$_{2.5}$-AMS equipped with a standard vaporizer (SV) was deployed once in China,

which provided new insights into composition and sources of PM$_{2.5}$ in Beijing and Xi'an (Elser et

al., 2016). The results showed that secondary inorganic components (mostly sulfate and nitrate)

and oxygenated organic aerosol (OOA) had large enhancements in large sizes (> 1 μm) during the

extreme haze periods in Beijing and Xi'an. It is clear that such real-time measurements of PM$_{2.5}$

compositions, particularly for a longer time with the new CV, in other polluted regions are

needed.





In this study, a $PM_{2.5}$-ACSM equipped with a CV was deployed for the first time in the megacity of Nanjing for the real-time measurements of NR-$PM_{2.5}$ compositions. The performance of the $PM_{2.5}$-ACSM is thoroughly evaluated by comparing with those measured by a suite of collocated on-line instruments, including a $PM_1$-ACSM, a Sunset Lab OC/EC Analyzer and a MARGA. The composition, diurnal variations, and processes of aerosol species in NR-$PM_1$ and NR-$PM_{1–2.5}$ are characterized and compared, moreover the sources of organic aerosols are elucidated by PMF. Finally, new insights into the impacts of aerosol liquid water on the formation of SIA and SOA are discussed in this study.

## 2 Experimental methods

All measurements took place from October 20 to November 16, 2015 in Nanjing, which is a typical mega-city in the western Yangtze River Delta of Eastern China. The sampling site is located at Jiangsu Environment Monitoring Center (32° 02′ 35″ N, 118° 44′ 45″ E), an urban station representative of an atmospheric environment subject to multiple source influences, including industry, traffic, cooking, and biogenic emissions, etc. More detailed descriptions of this sampling site can be found in previous studies (Zhang et al., 2015c; Zhang et al., 2015b; Zhang et al., 2017).

### 2.1 Instrumentation

In this study, two ACSMs, i.e., a $PM_1$-ACSM with SV and a $PM_{2.5}$-ACSM with CV were deployed side by side at the sampling site. The principles of the ACSM have been detailed elsewhere (Ng et al., 2011b). Briefly, ambient air is sampled into the aerodynamic lens system through a 100 μm diameter critical aperture with a flow rate of ~ 85 cc min⁻¹. The focused particle beam is transmitted through the differentially pumped vacuum chamber into the detection region. Aerosol particles impact and vaporize on an oven at the temperature of approximately 600



°C, and then are ionized with 70 eV electron impact. The produced ions are detected with a

quadrupole mass spectrometer (Ng et al., 2011b). Different from the AMS system, the

background of the ACSM is determined by measuring particle-free air.

The differences between the $PM_1$ and $PM_{2.5}$ ACSMs have been described in Xu et al.

(2017a). The three main modifications that enable accurate $PM_{2.5}$ quantification are the sampling

inlet, the aerodynamic lens, and the vaporizer. The sampling inlet of the $PM_{2.5}$-ACSM uses

straight flow paths and relatively short lengths of tubing to minimize particle loss. The particle

lens of the $PM_{2.5}$ ACSM operates at a higher pressure than that of the $PM_1$-ACSM (Liu et al.,

2007; Ng et al., 2011b) and transmits larger particles (Peck et al., 2016; Xu et al., 2017a). And the

standard vaporizer is replaced with the capture vaporizer to eliminate the effect of particle bounce

which can lead to a fraction of particle mass not being detected, an effect which increases at

larger particle diameters (Jayne et al., 2000; Hu et al., 2016; Xu et al., 2017a). The $PM_1$ and $PM_{2.5}$

ACSM mass spectra were analyzed using the ACSM Local toolkit (Version 1.5.11.2), a data

analysis software written in Wavemetrics Igor Pro™. The detailed procedures for the data

analysis have been described in Ng et al. (2011b) and Sun et al. (2012a). The response factors of

the two ACSMs were calibrated using the size-selected ammonium nitrate ($NH_4NO_3$) particle

(300 nm), which were $1.09 \times 10^{-10}$ and $2.06 \times 10^{-11}$, respectively for the $PM_1$ and the

$PM_{2.5}$-ACSM. The relative ionization efficiencies (RIEs) of ammonium and sulfate were

determined as 4.9 and 4.7, and 0.8 and 1.2 for the $PM_1$-ACSM and $PM_{2.5}$-ACSM, respectively.

The default RIE values of 1.1, 1.4, and 1.3 were used for nitrate, organics, and chloride,

respectively (Canagaratna et al., 2007; Ng et al., 2011b). In addition, the composition-dependent

CE, that is CE = max (0.45, 0.0833 + 0.9167 × ANMF) (Middlebrook et al., 2012), in which

ANMF is the mass fraction of ammonium nitrate, was used for the mass concentration

quantifications of the $PM_1$-ACSM species, while a CE = 1 was used for the $PM_{2.5}$-ACSM (Xu et

al., 2017a).

Water-soluble inorganic ions ($NH_4^+$, $Na^+$, $K^+$, $Ca^{2+}$, $Mg^{2+}$, $SO_4^{2-}$, $NO_3^-$, and $Cl^-$) in $PM_{2.5}$

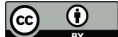



were simultaneously measured by a MARGA at 1 h resolution (Trebs et al., 2004; Rumsey et al., 2014). Ambient air was pulled into the MARGA sampling box with a flow rate of 16.7 L min$^{-1}$. After removing the interferences of water-soluble gases by a wet rotating denuder, aerosol

particles were dissolved into liquid phase, and then analyzed with two ion chromatographic systems (Metrohm USA, Inc., Riverview, FL, USA). In addition, the mass concentrations of OC and EC in PM$_{2.5}$ were measured using a Sunset Lab Semi-Continuous OC/EC Analyzer with true laser-based pyrolysis correction and compatibility with accepted NIOSH methods.

Particle number size distribution (3 nm–10 μm) was measured by a Twin Differential

Mobility Particle Sizer (TDMPS) combined with an Aerodynamic Particle Sizer (APS, TSI Model 3320). The TDMPS consists of two subsystems measuring different size ranges of dry particles at the same time. The 3–20 nm particles were measured by an Ultrafine Differential Mobility Analyzer in conjunction with an Ultrafine Condensation Particle Counter (TSI Model 3025) and the 20–900 nm particles were measured by a Differential Mobility Analyzer combined

with a Condensation Particle Counter (TSI Model 3010). Large particles between 900 nm and 10 μm were measured by the APS.

Other collocated measurements include the total PM$_1$ and PM$_{2.5}$ mass concentrations by a Met one BAM 1020 and a PM$_{2.5}$ Tapered Element Oscillating Microbalance equipped with a Filter Dynamic Measurement System (TEOM-FDMS, Thermo), respectively, and the gaseous

species of CO (model 48i), NO/NO$_2$ (model 42i), O$_3$ (model 49i), SO$_2$ (model 43i), and NH$_3$ (model 17i) by Thermo Scientific gas analysers. Meteorological parameters, including wind speed (WS), wind direction (WD), ambient temperature ($T$) and RH, and the parameters of solar radiation (SR) and precipitation were measured at the same sampling site.

## 2.2 ACSM data analysis

PMF analysis of the PM$_1$ and PM$_{2.5}$ ACSM organic mass spectra was performed within an Igor Pro-based PMF Evaluation Tool (Ulbrich et al., 2009) with PMF2.exe algorithm (Paatero and



Tapper, 1994). Pretreatment of the data and error matrices was similar to that reported in previous studies (Ulbrich et al., 2009; Zhang et al., 2011; Sun et al., 2012b). In addition, $m/z = 12$ and $m/z > 100$ were removed in the both ACSMs' organic PMF analysis considering that (1) a lot of

negative values at $m/z$ 12 due to background uncertainties; (2) small contribution of $m/z > 100$ in total organic signals (Ng et al., 2011b) and large uncertainties due to low ion transmission efficiency and interference from the internal standard naphthalene signals (Sun et al., 2012a). The PMF results were evaluated following the procedures detailed in Zhang et al. (2011). The detailed key diagnostic plots for the PMF solution of $PM_1$ and $PM_{2.5}$ ACSMs are shown in Figs. S1–S4 in

supporting information. For a better comparison, a simplistic PMF solution was used to extract two factors, a primary organic aerosol (POA) factor and a SOA factor for both $PM_1$ and $PM_{2.5}$ ACSMs. However higher order factor analysis utilizing PMF and multilinear engine (ME-2) (Canonaco et al., 2013) may reveal more chemical information which should be the subject of a future manuscript.

**2.3 Particle liquid water content (LWC)**

The LWC associated with inorganic species in $PM_1$ and $PM_{2.5}$ was predicted using ISORROPIA-II (Fountoukis and Nenes, 2007), respectively. The aerosol inorganic composition (measured by both $PM_1$ and $PM_{2.5}$ ACSMs, respectively) and meteorological parameters ($T$ and RH) were used as inputs and the ISORROPIA-II model then calculated the composition and

phase state of a $K^+–Ca^{2+}–Mg^{2+}–NH_4^+–Na^+–SO_4^{2-}–NO_3^-–Cl^-–H_2O$ in thermodynamic equilibrium with gas-phase precursors (Fountoukis and Nenes, 2007).

**2.4 Potential Source Contribution Function (PSCF) analysis**

Here, the 48 h back trajectories arriving at the sampling site at 100 m above ground level (a.g.l.) were calculated every 2 h using the Hybrid Single-Particle Lagrangian Integrated Trajectory

model (HYSPLIT, version 4.8) developed by National Oceanic and Atmospheric Administration




(NOAA) with GDAS meteorological field data (Draxler and Rolph, 2003). PSCF along with

calculated trajectories and air pollutants is a good approach to geographically identify the

potential contributions of regional sources to the receptor (Polissar et al., 1999). Each calculated

trajectory can be divided into some latitude–longitude coordinates corresponding to each grid cell

$(i, j)$. Briefly, the PSCF is calculated as:

$$PSCF\ (i, j) = (m_{ij\ /\ n_{ij}})\ w_{ij}$$

where $m_{ij}$ and $n_{ij}$ are the total number of back-trajectory segment endpoints that fall into the grid

cell, during all days and in days when receptor concentrations were higher than the criteria value,

respectively. In this study, the PSCF analysis performed using the ZeFir toolkit (Petit et al., 2017)

with a resolution of $0.2\,° \times 0.2\,°$ for each grid cell. And the 75th percentile of each aerosol species

during the entire study was used as the threshold value to calculate $m_{ij}$. In order to reduce the

influences of small $n_{ij}$ on the PSCF values, a weighting function ($w_{ij}$) proposed by a previous

study (Zeng and Hopke, 1989) was applied:

$$\omega_{ij} \ = \ \begin{cases} 1.00 & 80 < n_{ij} \\ 0.70 & 20 < n_{ij} \leq 80 \\ 0.42 & 10 < n_{ij} \leq 20 \\ 0.05 & n_{ij} < 10 \end{cases}$$

## 3 Results and discussion

### 3.1 Inter-comparisons

Figures 1 and 2 show the inter-comparisons of the measurements by the $PM_{2.5}$-ACSM with those

by other co-located instruments, including $PM_1$-ACSM, MARGA, OC/EC analyzer, and the total

$PM_{2.5}$ mass analyzers. Overall, the $PM_{2.5}$-ACSM measurements are well correlated with those

measured by co-located instruments ($r^2 > 0.9$), except for chloride. The NR-$PM_{2.5}$ concentrations

report approximately 90 % of the total $PM_{2.5}$ concentrations measured by the TEOM-FDMS

and/or BAM 1020 instruments (Fig. 2a). After considering the contributions of EC and alkaline





cations ($Na^+ + K^+ + Ca^{2+} + Mg^{2+}$), it can explain 92 % of the $PM_{2.5}$ mass. The slight underestimation of the total $PM_{2.5}$ mass might be primarily due to the un-identified mineral dust

components (e.g., Al, and Si) and sea salts (e.g., $Na^+$), and/or the measurement uncertainties between different techniques.

       SIA measured by the $PM_{2.5}$-ACSM were highly correlated with those measured by the MARGA ($r^2 = 0.92–0.95$). The absolute agreement between $PM_{2.5}$-ACSM and MARGA is very good for sulfate (slope = 1.02). The ammonium agreement is also quite good with the $PM_{2.5}$

ACSM measuring 89 % of that reported by the MARGA. The average ratios of the measured $NH_4^+$ to predicted $NH_4^+$ that requires to fully neutralize $SO_4^{2-}$, $NO_3^-$, and $Cl^-$ were 1.02 and 0.95 for the $PM_{2.5}$-ACSM and $PM_1$-ACSM, respectively (Fig. S5), which is similar to the water-soluble ion balance results from the MARGA (Fig. S6). For nitrate, however, the $PM_{2.5}$ ACSM measures about 68 % of what is reported by the MARGA. One reason might be due to the

contribution of nitrate from mineral dust and/or sea salt particles (e.g., $NaNO_3$ and $Mg(NO_3)_2$) (Gibson et al., 2006) that ACSM cannot detect due to the limited vaporizer temperature. However, the nitrate associated with $Na^+$ and $Mg^{2+}$ only accounts for 5 % of the total nitrate (Fig. S7a). Although the $PM_{2.5}$ ACSM and MARGA measurements both yielded a ~ 3:1 mass ratio of $NO_3^-$ to $NH_4^+$, consistent with that of $NH_4NO_3$, the exact reason for the ~ 30 % unexplained nitrate is

not clear yet. More future inter-comparison work is needed to better understand this. The much lower ratio of chloride (0.26, Figure 2f) between the $PM_{2.5}$-ACSM and MARGA suggests the presence of sea salt particles that are not expected to be vaporized in the ACSM, which only detects non-refractory ammonium chloride. For example, we estimate that 30 % of chloride existed in the form of NaCl (Fig. S7 c and d), which can explain a large fraction of the difference.

A recent evaluation of the AMS with a CV system also found a large difference in chloride measurement (Hu et al., 2017), yet the reason was not completely understood. A future RIE calibration for chloride in the CV system might be helpful to evaluate these differences.

       As seen in Figure 2b, organics measured by the $PM_1$ and $PM_{2.5}$ ACSMs both show good





correlations with the OC measured by the OC/EC Analyzer ($r^2 = 0.77$ and 0.93, respectively), but

the slopes, that is organic mass-to-organic carbon (OM / OC) ratio, were substantially different at

1.64 and 3.50, respectively. While the OM / OC ratio obtained from $PM_1$-ACSM dataset is

consistent with those (1.6–2.2) observed in previous studies, e.g., Pittsburgh (Zhang et al., 2005a)

and New York (Sun et al., 2011), such a high NR-$PM_{2.5}$ OM / OC ratio may appear surprising.

Indeed, most of previous studies generally reported ratio below 2.5 for aged OA (Aiken et al.,

2008; Zhang et al., 2011). However, the 3.5 value obtained here is close to values reported in few

other studies, e.g., a ratio of 3.3 observed in Pasadena (Hayes et al., 2013), and it may be

expected that the particles in NR-$PM_{1-2.5}$ are more aged and therefore have somewhat higher OM

/ OC ratios than those in NR-$PM_1$. Moreover, in the AMS and ACSM systems, the fraction of OA

signal at $m/z$ 44 ($f_{44}$), mostly dominated by $CO_2^+$, is commonly considered as a surrogate of

atomic oxygen-to-carbon (O / C) and OM / OC (Aiken et al., 2008; Ng et al., 2011a). As reported

from the ACTRIS ACSM inter-comparison works, instrument artefacts may significantly affect

the variability in $f_{44}$ measured by different ACSMs (Crenn et al., 2015). For example, Pieber et al.

(2016) recently found that thermal decomposition products of inorganic salts on the SV may raise

up non-OA $CO_2^+$ signal, which can increase $f_{44}$ values. Therefore, the impact of instrument

artefacts on the $PM_{2.5}$ ACSM should be also investigated in future study. Another reason for this

discrepancy is likely that OC is underestimated by the Sunset OC/EC analyzer due to evaporative

loss of semi-volatile organic carbon during the sampling. It is also possible that large particles are

not being efficiently delivered to the filter in the semi-continuous OC/EC analyzer as they pass

through a warm solenoid valve with a bent flow path upstream of the filter.

Figure 2 also shows that SIA measured by the $PM_1$-ACSM were tightly correlated with

those by the MARGA ($r^2 = 0.68$–0.87), indicating that the temporal variations of inorganic

species in NR-$PM_1$ are generally similar to those in $PM_{2.5}$. However, the SIA in NR-$PM_1$ only

report 25–49 % of those in $PM_{2.5}$, indicating that a large fraction of SIA is present in the size

range of 1–2.5 μm (NR-$PM_{1-2.5}$). As shown in Fig. S8b, the average ratio of NR-$PM_1$ to





NR-PM$_{2.5}$ is 0.48, suggesting that nearly half of NR-PM$_{2.5}$ is NR-PM$_{1–2.5}$. This is quite different

from the results observed in US and Europe that a dominant fraction of PM$_{2.5}$ is PM$_1$ (Sun et al.,

2011; Petit et al., 2015). For instance, 91 % of PM$_{2.5}$ nitrate was found in NR-PM$_1$ at an urban

background site in Paris, France (Petit et al., 2015). Our results indicate that it is of great

importance to chemically characterize PM$_{1–2.5}$ in China because of their large contributions to the

total mass of PM$_{2.5}$ in accordance with Elser et al. (2016).

## 3.2 Sized-segregated investigation of NR-PM$_{2.5}$ components

Figure 3 presents the time series of the mass concentrations of the NR-PM$_1$ and NR-PM$_{2.5}$ species,

meteorological parameters, gas-phase species and size-resolved particle number concentrations

for the entire study. The entire study period was characterized by five episodes (Ep1–Ep5)

according to different pollution events as marked in Fig. 3e. The mass concentrations of the total

NR-PM$_1$ and NR-PM$_{2.5}$ vary dramatically throughout the entire study, ranging from 4.2 to 81.9

µg m$^{-3}$, and 9.3 to 178.7 µg m$^{-3}$, respectively. For example, aerosol mass loadings increase

rapidly from a few µg m$^{-3}$ to hundreds of µg m$^{-3}$ within a short-time scale, e.g., Ep2, Ep4, and

Ep5, which are associated with new particle formation and growth (Ep2) and foggy days (Ep4

and Ep5), respectively (Fig. 3e). We also noticed that such rapid changes in aerosol mass were

generally associated with a wind direction change to the northwest (Fig. 3a). This result indicates

the potential source contributions in the northwest region to the PM level in Nanjing. The average

NR-PM$_1$ and NR-PM$_{2.5}$ were 32.5 µg m$^{-3}$ and 68.7 µg m$^{-3}$, respectively, for the entire study,

indicating that 53 % of PM$_{2.5}$ mass is in the size range of 1–2.5 µm. During the persistent

pollution events, e.g., Ep1 and Ep2, NR-PM$_{1–2.5}$ accounts for 56 % and 42 % of the total

NR-PM$_{2.5}$. Overall, NR-PM$_{1–2.5}$ also shows a ubiquitously higher contribution to NR-PM$_{2.5}$ than





that of NR-PM$_1$ during different types of episodes, except Ep3, further highlighting the importance for characterization of aerosol particles between 1 and 2.5 μm.

### 3.2.1 Secondary inorganic aerosols

SIA constitutes a major fraction of NR-PM$_{2.5}$, on average accounting for 61 % in this study (Fig. 4). The average mass concentrations of SIA in NR-PM$_1$ and NR-PM$_{2.5}$ were 19.6 μg m$^{-3}$ and 40.6 μg m$^{-3}$, respectively, both of which are about 1.6–1.7 times higher than that of organics. The average mass concentration of sulfate in NR-PM$_1$ is 5.9 μg m$^{-3}$, which is close to that (5.4 μg m$^{-3}$) measured by a soot particle (SP) AMS during springtime in urban Nanjing (Wang et al., 2016b).

However, it is nearly 3 times lower than that in NR-PM$_{2.5}$ (17.4 μg m$^{-3}$) indicating that a major fraction of sulfate exists in the size range of 1–2.5 μm. Sulfate frequently comprises the largest fraction of NR-PM$_{1–2.5}$ with SOA being the second largest, particularly in the polluted episodes (Fig 4b). On average, sulfate and SOA contribute 33 % and 30 % to the total NR-PM$_{1–2.5}$, respectively, during the entire periods. Sulfate accounts for the largest contribution (41 %) to the

total NR-PM$_{1–2.5}$ loadings during the persistent pollution event (Ep1). Compared with sulfate (26 %), nitrate accounts for a lower fraction (19 %) of NR-PM$_{2.5}$ for the entire study, and the contributions to NR-PM$_{1–2.5}$ is typically 2–4 times lower than that in NR-PM$_1$. One reason is likely that non-refractory nitrate (e.g., ammonium nitrate) mainly existed in submicron aerosols, while those in NR-PM$_{1–2.5}$ contains more nitrate from sea salt and mineral dusts.

### 3.2.2 POA and SOA

Figure 5a shows a comparison of the mass spectra of POA and SOA between NR-PM$_1$ and NR-PM$_{2.5}$. While the mass spectra were overall similar, the one resolved from the PM$_{2.5}$-ACSM with capture vaporizer showed higher contributions of small *m/z*'s. This is consistent with the recent findings that the CV is subject to have enhanced thermal decomposition compared to the



SV (Hu et al., 2016). Similar to previous studies, the POA spectrum is characterized by typical

hydrocarbon ion series $C_nH_{2n-1}^+$ and $C_nH_{2n+1}^+$ (Zhang et al., 2011), e.g., $m/z$ 55 and $m/z$ 57, as well

as AMS biomass-burning tracers (Alfarra et al., 2007), e.g., $m/z$ 60 and $m/z$ 73. Note that the mass

spectra of NR-PM$_{2.5}$ shows smaller fractions of $m/z$ 60 and $m/z$ 73 signals, compared with those

of PM$_1$, which is likely due to the stronger thermal decomposition (Pieber et al., 2016). The high

ratio of $m/z$ 55/57 in the SV system suggests a significant influence from local cooking emissions

(Allan et al., 2010; Mohr et al., 2012; Sun et al., 2012a; Zhang et al., 2015c). In addition to the

noon and evening meal time peaks, the diurnal variations of POA in Fig. S9 also show two peaks

corresponding to morning rush hours (Zhang et al., 2015b), and night biomass-burning emissions

(Zhang et al., 2015c). This result suggests that the POA factor in this study is subject to multiple

influences, including traffic, cooking, and biomass burning emissions. The mass spectrum of

SOA in both NR-PM$_1$ and NR-PM$_{2.5}$ are dominated by $m/z$ 44 (mostly $CO_2^+$) with a higher $f_{44}$ in

the NR-PM$_{2.5}$ system. One reason for the higher $f_{44}$ in the PM$_{2.5}$-ACSM could be due to the effects

of enhanced thermal decomposition in the CV system (Xu et al., 2017a). Another possibility is

the more crustal materials in PM$_{1-2.5}$ which can produce non-OA $CO_2^+$ interference signal from

the reactions on the particle SV (Pieber et al., 2016; Bozzetti et al., 2017). For example, the

deposited carbonates on the particle vaporizer in AMS/ACSM system may release $CO_2^+$ signal

upon reaction with HNO$_3$ and NO$_x$ (Goodman et al., 2000; Pieber et al., 2016). In addition, as

discussed in Sect. 3.1, the instrument artefacts may lead to the $f_{44}$ discrepancies among different

ACSM instruments and thereby affect factor profiles in ME-2/PMF analysis (Fröhlich et al.,

2015), which might also be the potential impact on the PMF analysis of PM$_{2.5}$-ACSM OA mass

spectra in this study.



The average mass concentration of OA in NR-PM$_{2.5}$ (25.2 μg m$^{-3}$) is approximately twice that in NR-PM$_1$ (11.3 μg m$^{-3}$). Despite the large differences in mass concentrations, the contributions of organics to the total NR-PM$_{1-2.5}$ and NR-PM$_1$ are relatively similar (40 % vs.

36 %). POA on average contributes 34 % to the total OA in NR-PM$_1$, which is higher than the contribution (28 %) in NR-PM$_{2.5}$ during the entire study. In contrast, SOA showed a higher fraction in OA in NR-PM$_{2.5}$ (72 %) than NR-PM$_1$ (66 %). As shown in Fig. 5, the mass concentrations (9.0–11.8 μg m$^{-3}$) and mass fractions (14–20 %) of SOA in NR-PM$_{1-2.5}$ are also ubiquitously higher than those in NR-PM$_1$ (4.3–10.4 μg m$^{-3}$, and 10–13 %).

**3.3 Effects of aqueous and photochemical processing on secondary aerosol formation**

Figure 6 shows positive relationships between the sum of sulfate and nitrate and the molar ratio of [NH$_4^+$] to [NH$_3$ + NH$_4^+$], a proxy for the NH$_3$ gas-to-particle conversion ratio, in PM$_1$ and PM$_{2.5}$, and the ratio of [SO$_4^{2-}$ + NO$_3^-$] / [NH$_4^+$ / (NH$_3$ + NH$_4^+$)] increased as a function of RH.

Atmospheric NH$_3$ can react rapidly with H$_2$SO$_4$ or HNO$_3$ to form secondary inorganic salts (Wang et al., 2015a; Zhang et al., 2015a). Recent studies showed that sulfate formation was more sensitive to aqueous oxidation of SO$_2$ in the presence of abundant NO$_x$ and high RH levels during the haze pollution periods in China, while the role of gas-phase oxidation processing is comparably small (Sun et al., 2013; Xie et al., 2015; Cheng et al., 2016; Wang et al., 2016a; Xue

et al., 2016). The aqueous oxidation of SO$_2$ by NO$_2$ chemistry pathway could be e favored in our study due to the NH$_3$-rich environment (Wang et al., 2016a), supporting that the sulfate and nitrate formation under high RH conditions was likely enhanced here by rapid combination with



$NH_3$. In addition, we found that the number concentrations of accumulation mode (0.1–1 μm) and medium-size mode (1–2.5 μm) particles show obvious enhancement as the molar ratio of $[NH_4^+]$

to $[NH_3 + NH_4^+]$ increases, yet Aitken mode particles (0.02–0.1 μm) show a decreasing trend (Fig. 7a). Results here might suggest that the small particles formed through gas-phase $NH_3$ conversation can rapidly grow to large particles in $NH_3$-rich and high RH environments, which is also consistent with the large increase of the total particle surface area as a function of $[NH_4^+]$ / $[NH_3 + NH_4^+]$ and aerosol water driven by SIA (Fig. 7b).

SOA shows a positive relationship with aerosol liquid water (Fig. 8a), and the slope ratio of SOA to LWC is strongly dependent on RH levels. For example, the ratios at low RH levels (RH < 40 %) (2.25 and 2.50 in $PM_1$ and $PM_{2.5}$, respectively) are much higher than those at high RH levels (RH > 80 %, slope = 0.18 and 0.22). These results indicate the different roles of atmospheric processing in SOA formation at different RH levels. As shown in Fig. 8b, SOA

correlates well with $[SO_4^{2-} + NO_3^-]$ ($r^2 = 0.72$ and 0.75 for NR-$PM_1$ and NR-$PM_{2.5}$, respectively), and the correlation coefficient shows an evident RH dependence with a stronger correlation at high RH levels (e.g., RH > 80 %, $r^2 = 0.92$). This suggests that SOA might be well internally mixed with SIA, and the enhancement of SOA might be caused by aqueous-phase chemistry under high RH levels in urban Nanjing. In addition, the ratio of SOA to $[SO_4^{2-} + NO_3^-]$ is also

dependent on RH, with higher slopes (0.58 and 0.75 for NR-$PM_1$ and NR-$PM_{2.5}$, respectively) at RH < 40 % and lower values at RH > 80 (0.41 and 0.50, respectively), suggesting that the enhancement of SIA was higher than the SOA production via aqueous-phase chemistry pathways. High SOA at low RH levels were likely mainly from photochemical production, which is also supported by the correspondingly high $O_x$ (= $O_3 + NO_2$) levels (Fig. 9). Figure 9 also shows that





the SOA concentrations in both $PM_1$ and $PM_{2.5}$ increase as the increases of $O_x$, and the ratios of

SOA to $O_x$ show clear enhancements as the increases of RH levels. For example, the ratio of

[NR-$PM_1$ SOA] / [$O_x$] at low RH conditions (RH < 50 %) is close to that observed in our

previous study during period with strong photochemical processing (Zhang et al., 2017). The

mass spectra of OA are also substantially different between low and high RH and/or $O_x$ levels

(Fig. S10). For instance, the mass spectra of SOA in both NR-$PM_1$ and NR-$PM_{2.5}$ were

characterized by higher signals at $m/z$ 44 at high RH levels, likely suggesting the formation of

more oxidized SOA via aqueous processing (Xu et al., 2017b). These results might indicate that

the total SOA contains different types of SOA at low and high RH levels. While the formation of

SOA at high RH levels is significantly affected by aqueous-phase processing, it might be driven

more by photochemical processing at low RH levels.

      Considering the atmospheric dilution effects, the OA formation and evolution processes can

be evaluated using the OA / ΔCO ratios (ΔCO is the CO minus background CO) (Dunlea et al.,

2009; DeCarlo et al., 2010). A background CO of 0.02 mg m$^{-3}$ was calculated as the lowest 5 %

of data during the entire study. Figure 10 shows OA versus ΔCO for the entire study, and a

comparison with those from previous studies (Zhang et al., 2005b; Aiken et al., 2009; de Gouw

and Jimenez, 2009; Hu et al., 2013; Sun et al., 2014; Zhang et al., 2017). The POA / ΔCO in this

study is overall consistent with HOA / ΔCO observed at other urban cities, suggesting that the

POA in this study might be dominated by urban traffic emissions. The high OA / ΔCO ratios in

NR-$PM_1$ and NR-$PM_{2.5}$ likely indicate important impacts of SOA formation and/or biomass

burning sources on OA burden in urban Nanjing. Figure S11 shows similar increases in gaseous

species (including $NO_2$, $SO_2$ and $NH_3$) to those of SOA and the total particle surface area,



respectively. Again, similar increases in $NO_2$, $SO_2$, and $NH_3$ as a function of surface area were also observed (Fig. S11 d–f). The ratio of [SOA] / [ΔCO], a proxy for the SOA production rate, also increases as the total particle surface area and aerosol liquid water increase (Fig. 11). A

recent study found that aqueous-phase reaction of $NO_2$ with $SO_2$ not only produces sulfate but also nitrite (Cheng et al., 2016), which may become hygroscopic leading to high aerosol water levels under high RH conditions (Wang et al., 2016a; Chu et al., 2016). Therefore, aqueous $SO_2$ oxidation by $NO_2$ efficiently produces SIA with the $NH_3$ neutralization (Wang et al., 2016a), which may further increase SOA productions under high RH levels in this study. Our results

highlight that $NO_2$ reaction with $SO_2$ in aerosol water may not only produce SIA, but also can enhance SOA productions via aqueous-phase chemistry pathways under high RH conditions during the haze episodes.

### 3.4 Specific episodes analysis (Ep2 and Ep5)

Figure 12 shows the temporal variations of secondary aerosols, including SOA and SIA, in

NR-PM$_1$ and NR-PM$_{1-2.5}$ during two different episodes. A clear particle nucleation and growth event was observed before the formation of the first episode (Ep2, Fig. 12a), during which the air was relatively clean (PM$_{2.5}$ mass loading = 28.5 μg m$^{-3}$) and SR was strong (610.5 W m$^{-2}$). The number concentration of nucleation mode particles increased rapidly from ~ 670 to 2400 (# cm$^{-3}$) within 1 hour, and the particle size grew from ~ 3 nm to 100 nm during the rest time of the day.

The role of new particle formation and growth in the formation of haze pollution has been reported in urban environments (Guo et al., 2014). Here, we observed simultaneous increases in secondary aerosol species. Given the relatively stagnant conditions (WS = 1 m s$^{-1}$, on average),



and the simultaneous increases in gaseous $NH_3$ and $SO_2$ during the particle growth period.

Comparatively, $NO_x$ shows a pronounced night peak and then decreases rapidly during daytime

because it is mainly from local traffic emissions. Although only one such case was observed

throughout the entire study due to the suppression of new particle formation by abundant

preexisting particles under the polluted environments, it appears that the continuous growth from

nucleation mode particles under abundant $NH_3$, $SO_2$, and $NO_x$ might also be one of the reasons

for the high PM pollution in Nanjing.

455        The formation of secondary aerosol was more rapid during Ep5 compared to Ep2 (Fig. 12b),

and was clearly associated with a fog event (RH > 80 % and averaged LWC = 53.9 μg m$^{-3}$).

While the number concentrations of Aitken mode particles remained small, the mass

concentrations of secondary sulfate, nitrate and SOA show dramatic increases along with

simultaneous increases in large particles ($D_m$ > 100 nm) (Fig. 12b). This is likely due to the

efficient uptake kinetics of gaseous species (e.g., $SO_2$ and $NO_2$) upon preexisting aerosol water

(Cheng et al., 2016; Xue et al., 2016), which may undergo aqueous/heterogeneous reactions and

subsequent hygroscopic growth at high RH. In fact, the mass fractions of secondary species of

NR-PM$_{1-2.5}$ in PM$_{2.5}$ increased from 33 % to 56 %. These results support that aqueous processing

play a more important role in haze formation under high RH conditions and it tends to form more

large particles. The enhancement of SOA production via aqueous-phase chemistry has been

observed in many previous field studies (Ge et al., 2012; Chakraborty et al., 2015; Sun et al.,

2016). As discussed above, SOA in this study shows a good correlation with [$SO_4^{2-}$ + $NO_3^-$] and

particle water (under high RH levels), indicating that aqueous chemistry during foggy days might

facilitate the production of both SIA and SOA (Xie et al., 2015; Wang et al., 2016a; Chu et al.,



2016). This is also consistent with previous results observed during haze events in several urban

regions in China (Wang et al., 2016a). We also compared the OA mass spectra between the two

episodes. The OA mass spectra during the fog episode were characterized by much higher $m/z$ 44

and $f_{44}$ compared with that during the new particle formation episode (Fig. S10). This result

indicates different formation mechanisms of SOA between the two different episodes.

Chakraborty et al. (2015) have also observed similar aerosol composition differences between

foggy and non-foggy events with a high-resolution aerosol mass spectrometer instrument

deployed in Kanpur, India. While photochemical processing is the major formation mechanism of

Ep2, aqueous-phase processing is more important for the formation of more aged SOA.

## 3.5 Geographic origins

The potential source regions of SIA and SOA were investigated with PSCF. As shown in Fig. 13,

SOA and SIA all showed high potential source regions mainly located in the east and west with

high anthropogenic-rich emissions in the YRD region (Fig. S12). These results indicate that

secondary aerosols are mainly formed over a regional scale in the west and east regions. However,

we also found a clear regional transport dominantly from the north of Nanjing during the

formation stage of Ep5 (Fig. S13). These results suggest that secondary aerosol from regional

transport associated with the northern winds can also play a dominant role in the formation of

specific haze events in YRD region of Eastern China. In contrast, POA shows a dominant

potential source region located near the sampling site, supporting the idea that primary aerosols

were mainly emitted in a local scale.



## 4 Conclusions and Implications


The chemically-resolved mass concentrations of NR-PM$_{2.5}$ were measured in-situ by the newly developed PM$_{2.5}$-ACSM in urban Nanjing, China for the first time. The measured NR-PM$_{2.5}$ chemical species (organics, sulfate, ammonium, and nitrate) correlated well ($r^2 > 0.9$) with those from co-located measurements by the MARGA and OC/EC Analyzer. Also, all NR-PM$_{2.5}$ species

were tightly correlated with those in NR-PM$_1$ that were measured by a PM$_1$-ACSM. The comparisons between the two different ACSMs revealed substantial mass fractions of aerosol species in NR-PM$_{1–2.5}$, yet the ratios of [NR-PM$_1$] / [NR-PM$_{2.5}$] varied among different species. In particular, nitrate and chloride showed much higher [NR-PM$_1$] / [NR-PM$_{2.5}$] ratios compared with other species. The reasons are not very clear yet although refractory mineral dust and sea

salts can explain some differences. PMF analysis also showed similar temporal variations in POA and SOA between NR-PM$_1$ and NR-PM$_{2.5}$, but the mass spectra were slightly different with higher $f_{44}$ and more small fragments for OA in NR-PM$_{2.5}$ due to enhanced thermo decomposition.

On average, NR-PM$_{2.5}$ was mainly composed of SOA (27 %) and SIA (61 %) for the entire study, of which 16 % of SOA and 17 % of sulfate presented in the size range of 1–2.5 μm. Sulfate

and nitrate showed a positive relationship with the molar ratio of [NH$_4^+$] / [NH$_3$ + NH$_4^+$], revealing the enhancement of SIA productions in such NH$_3$-rich environments. A positive relationship between SOA and aerosol liquid water, and simultaneous enhancements in the ratio of [SOA] / [ΔCO] and gas-phase species (NO$_2$, SO$_2$, and NH$_3$) loadings as a function of particle surface area and aerosol water were observed. These results suggest that the increased aqueous

aerosol surface may enhance SOA production via heterogeneous reactions. Therefore, decreasing anthropogenic NO$_2$, SO$_2$, and NH$_3$ emissions may reduce both SIA and SOA levels. Further analysis showed that the high potential source regions of secondary aerosols are mainly located in the west and east although regional transport from the polluted northern region can affect specific haze events as well.



515        Furthermore, episodes analysis showed that secondary aerosol species (SIA and SOA) in

NR-PM$_{1-2.5}$ showed rapid increases within several hours during the fog processing which also

contributed the dominant fractions of the total PM$_{2.5}$ mass while smaller particles (less than 100

nm) remained relative unchanged, indicating an enhanced role of aerosol species in PM$_{1-2.5}$

during the fog episode. In contrast, photochemical processing played a more important role in

driving aerosol variations during the new particle formation and growth event. Overall, our study

highlights the importance of real-time characterization of PM$_{2.5}$ compositions to study the sources

and processes of fine particles in China.



## Acknowledgments

This work was supported by Natural Science Foundation of China (D0512/91544231) and the

National Key Research and Development Plan of China (2016 YFC0200505). The development

of the $PM_{2.5}$-ACSM was funded by US EPA grant # EP-D-12-007 and US DoE grant #

DE-SC0001673. We would like to thank Dr. Ping Chen for his supports on this campaign.

Yunjiang Zhang acknowledges the PhD Scholarship from the China Scholarship Council (CSC).




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






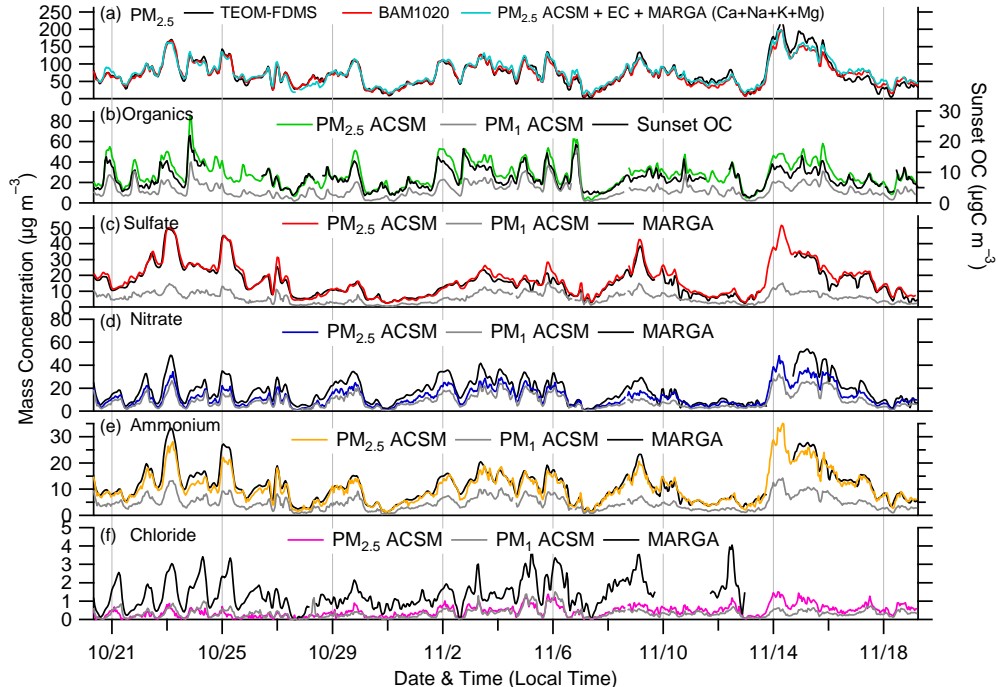

**Figure 1.** Inter-comparisons between the NR-PM$_{2.5}$ mass concentrations measured by the PM$_1$

and PM$_{2.5}$ ACSMs and the data acquired by collocated instruments: (a) NR-PM$_{2.5}$ vs. PM$_{2.5}$ mass

by a TEOM and a MET ONE BAM-1020, (b) organics vs. PM$_{2.5}$ OC by a Sunset Lab OC/EC

Analyzer, and (c–f) sulfate, nitrate, ammonium, and chloride vs. those measured by the PM$_{2.5}$

MARGA.





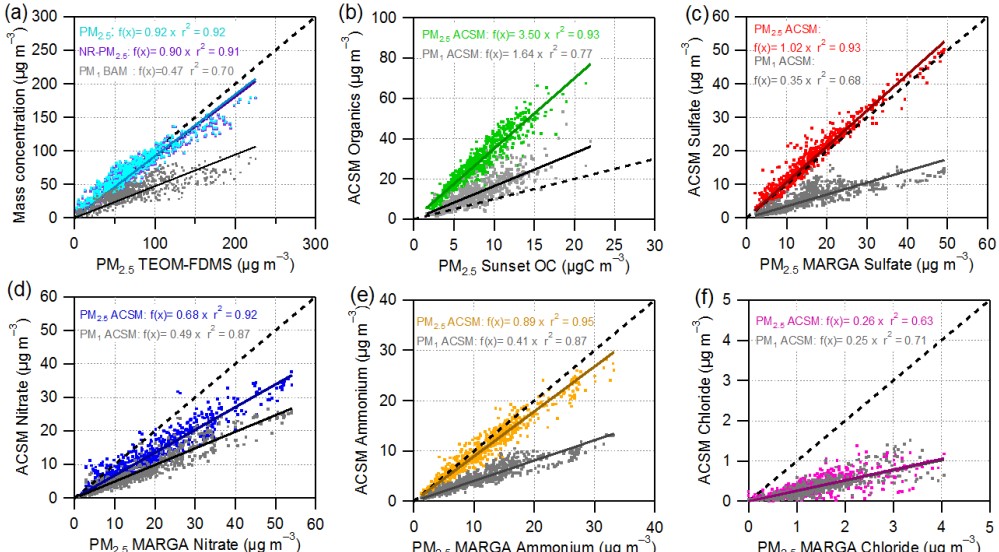

**Figure 2.** Scatter plots with the linear regression parameters and the 1:1 line (dash line) shown

for the comparisons. Note that the term of "$PM_{2.5}$" in the plot of Fig. 2a means that the sum mass

concentration of $PM_{2.5}$-ACSM species (organics, nitrate, sulfate, ammonium, and chloride),

Sunset EC, and MARGA species ($K^+$, $Na^+$, $Mg^{2+}$, and $Ca^{2+}$).





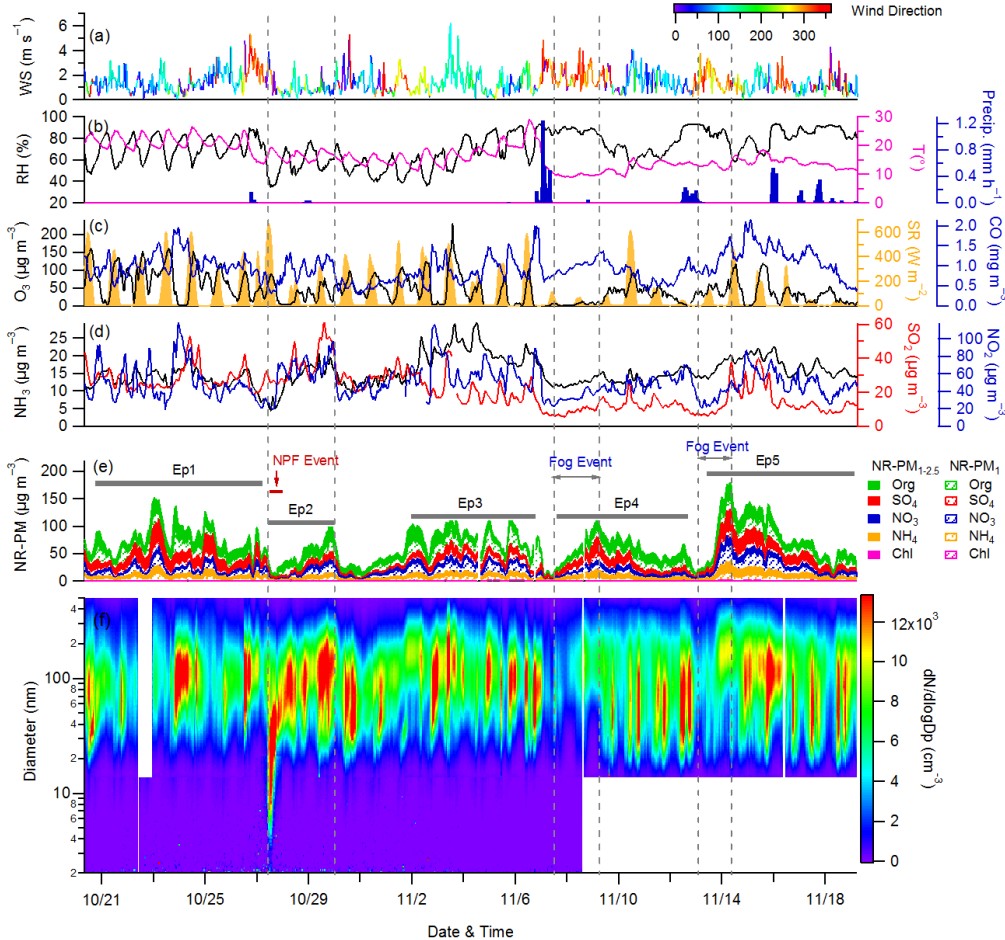

**Figure 3.** Time series of (a) wind direction (WD) and wind speed (WS); (b) relative humidity (RH), air temperature ($T$); (c) solar radiation (SR) and $O_3$; (d) gas-phase $NH_3$, $SO_2$ and $NO_2$; (e) chemical composition of NR-PM ($PM_1$ and $PM_{1-2.5}$); and (f) size distribution of aerosol particles during the entire study. Note that the white blank areas in the (f) are coursed by the missing data. In addition, five episodes (Ep1–Ep5) are marked by different pollution events, e.g., persistent haze pollution ($> \sim 5$ days) (Ep1 and Ep3), new particle formation and growth evolution (Ep2), and fog related processes (Ep4 and Ep5).





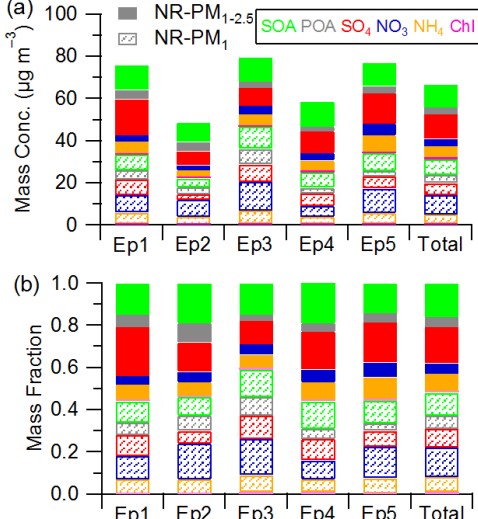

**Figure 4.** Mass concentration (a) and fraction (b) of NR-PM$_1$ and NR-PM$_{1-2.5}$ chemical

components in NR-PM$_{2.5}$ respectively during different episodes (Ep1–Ep5) marked in Figure 3

and entire study period (Total).




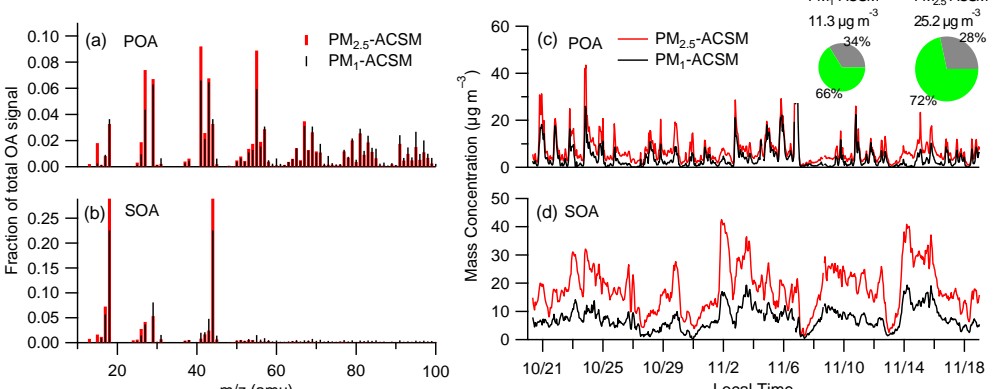

**Figure 5.** Mass spectra (a and b) and time series (c and d) of POA and SOA for the $PM_1$-ACSM

and $PM_{2.5}$-ACSM, respectively. The average mass concentrations and fraction of POA and SOA

were added in the sub-plots.



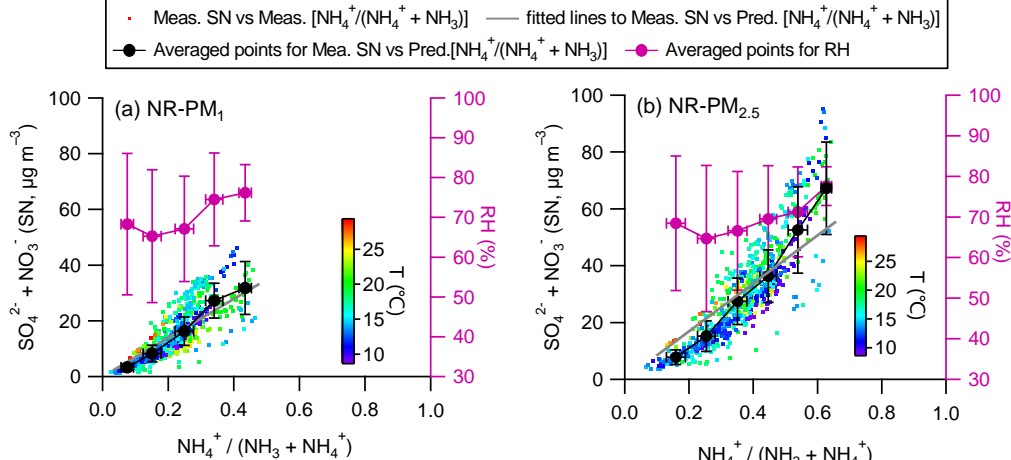

**Figure 6.** Relationship between the ratio of $[NH_4^+]$ / $[NH_3+NH_4^+]$ and the sum of $[SO_4^{2-} + NO_3^-]$,

as well as SOA in (a) NR-PM$_1$ and (b) NR-PM$_{2.5}$ respectively. Note that predicted (Pred.) $[NH_4^+]$

/ $[NH_3 + NH_4^+]$ is calculated by Pred. $NH_4^+$ (estimated by the fully neutralized $NO_3^-$ and $SO_4^{2-}$)

and measured (Meas.) $NO_3^-$ and $SO_4^{2-}$ from the PM$_1$-ACSM and PM$_{2.5}$-ACSM, respectively.



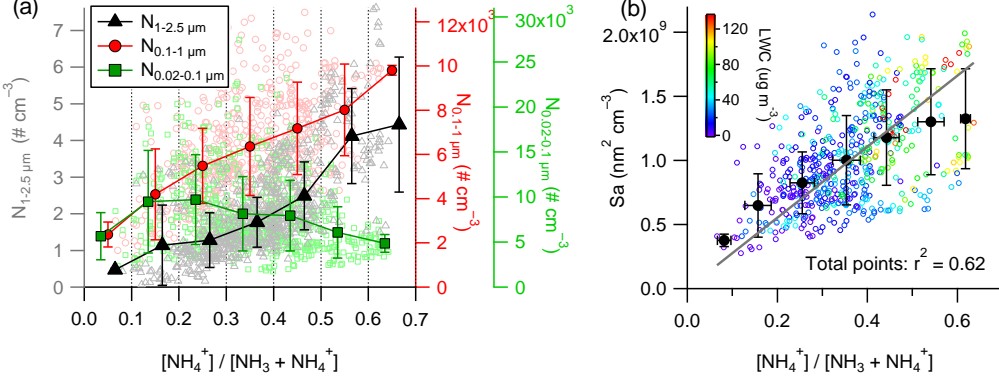

**Figure 7.** Relationships between the ratio of $[NH_4^+]$ / $[NH_3 + NH_4^+]$ and (a) the number concentrations at different mode particles, i.e., Aiken mode (0.02–0.1 μm), Accumulation mode (0.1–1 μm) and medium-size mode (1–2.5 μm), and (b) total aerosol surface area colored by aerosol water driven by secondary inorganic aerosols in NR-PM$_{2.5}$.




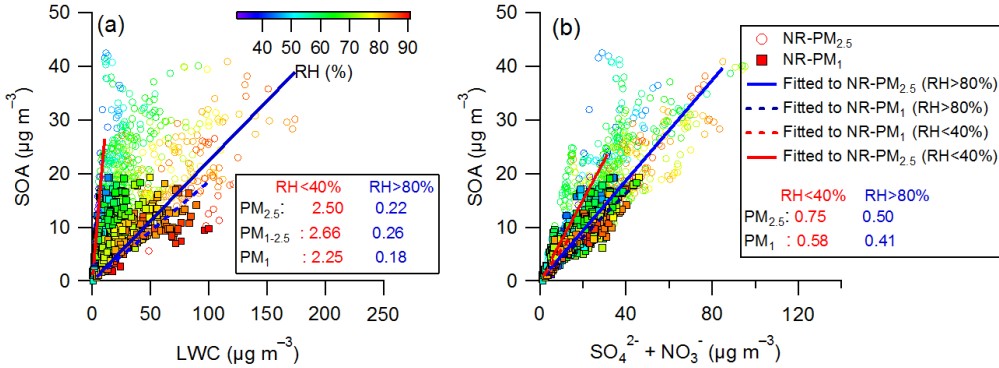


**Figure 8.** Correlations between (a) SOA versus LWC and (b) SOA versus $SO_4^{2-} + NO_3^-$, which is color coded by RH. The regression slopes at different RH levels (RH < 40 % and RH > 80 %) and in different size ($PM_1$ and $PM_{2.5}$) are also shown. Note that the wet scavenging particles were removed in this calculation.






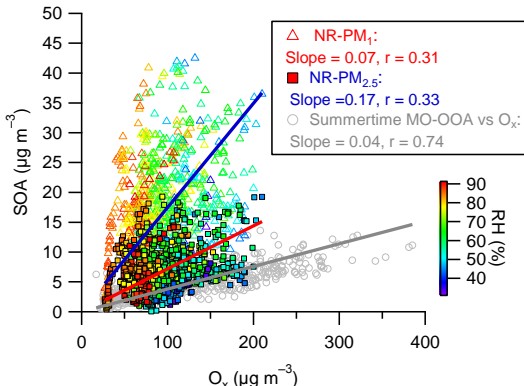

**Figure 9.** Relationship between NR-PM$_1$ and NR-PM$_{2.5}$ SOA and O$_x$ (= O$_3$ + NO$_2$) respectively.
Note that more oxidized OOA (MO-OOA) was observed at the same sampling site during
summertime (August) 2013 (Zhang et al., 2017).






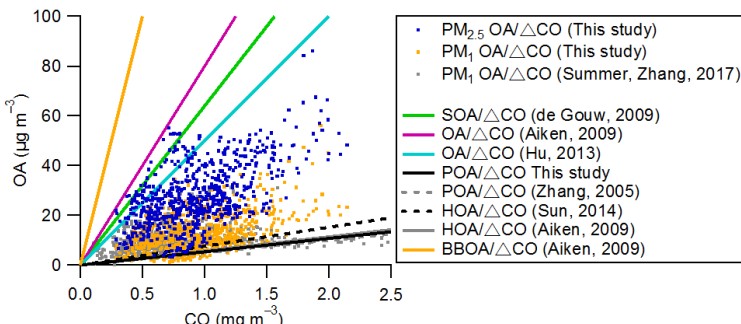

**Figure 10.** Scatter plot of NR-PM$_1$ and NR-PM$_{2.5}$ OA vs. CO during the entire study. The reference lines in the plot are from previous studies (Zhang et al., 2005b; Aiken et al., 2009; de

Gouw and Jimenez, 2009; Hu et al., 2013; Sun et al., 2014; Zhang et al., 2017).





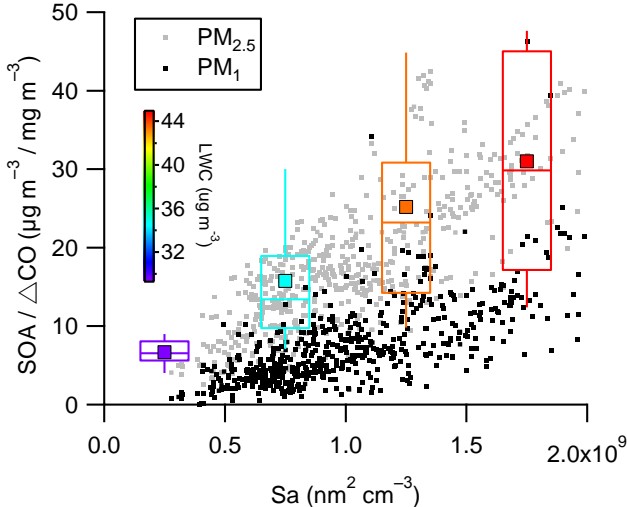

**Figure 11.** Relationship between the ratio of [SOA] / [ΔCO] and total particle surface area ($S_a$) colored by LWC in NR-PM$_{2.5}$. The NR-PM$_{2.5}$ SOA data are only binned here according to $S_a$. Note that background CO of 0.02 mg m$^{-3}$ was calculated as the lowest 5 % of data during the entire study.




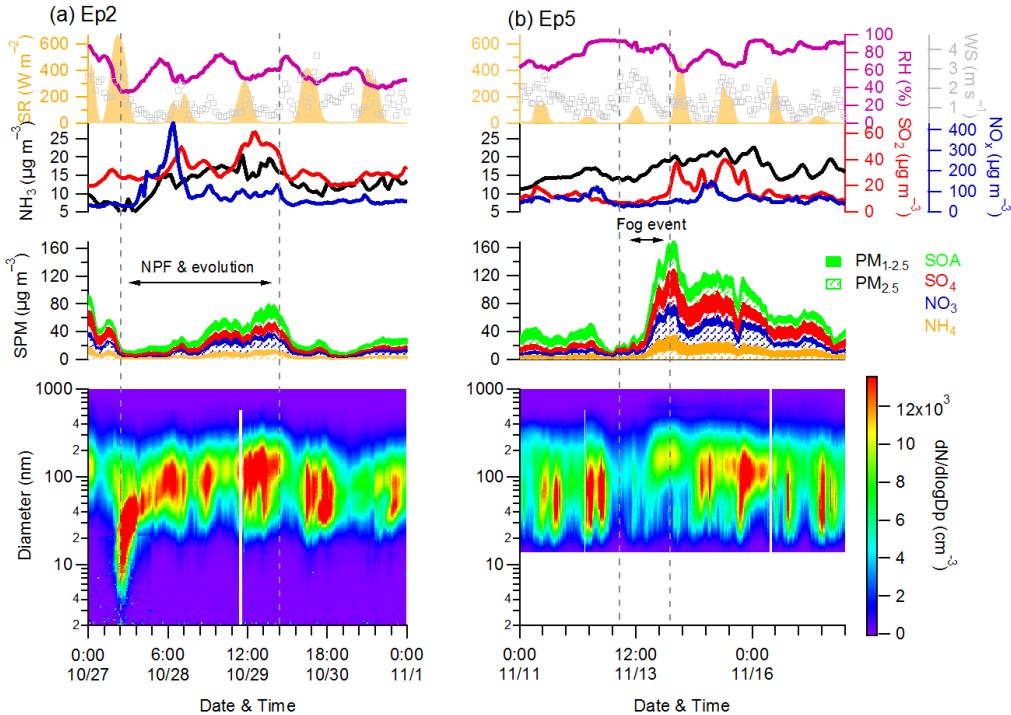

**Figure 12.** Evolution of meteorological parameters, secondary particulate matter (SPM), and size

distribution during the two types of episodes (Ep2 and Ep5).





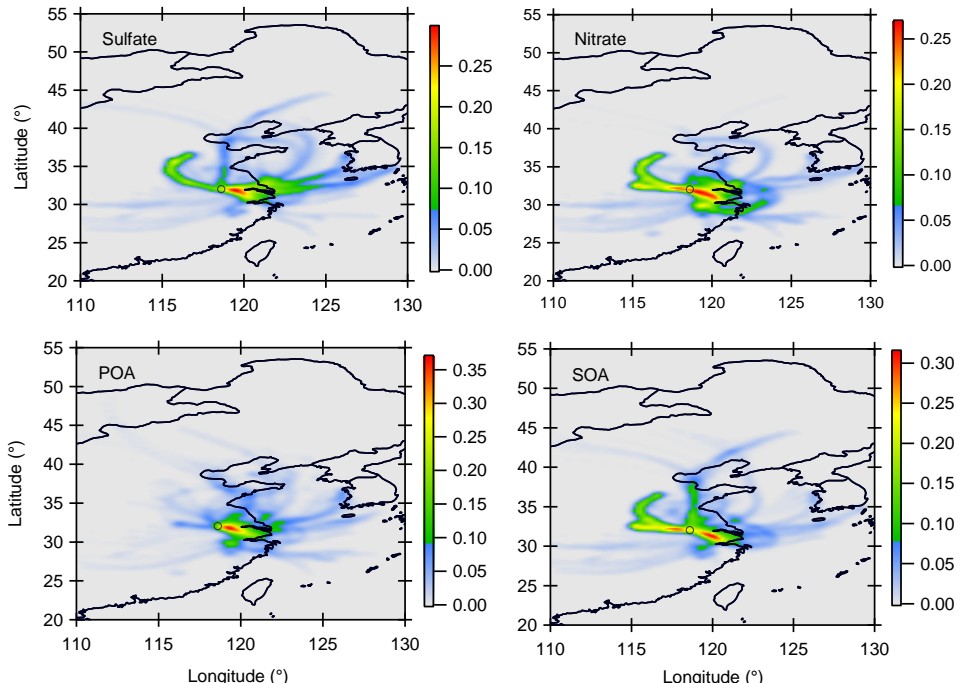

**Figure 13**. PSCF analysis of the NR-PM$_{2.5}$ secondary inorganic aerosol (nitrate and sulfate) and

organic components (POA and SOA).