# Peer review of "Field characterization of the $PM_{2.5}$ Aerosol Chemical Speciation Monitor: insights into the composition, sources and processes of fine particles in Eastern China"

_Atmospheric Chemistry and Physics, 2017_

## Referee Comment (RC1) · Anonymous Referee #1 · 1 Jun 2017

This data shows a comparison of ACSM data in Nanjing between a traditional PM1 instrument and a new PM2.5 instrument equipped with a capture vaporiser (CV), along with other instruments. These two modifications to the instrument design have been heavily anticipated for the purposes of comparability with other PM2.5 measurements and to get away from the 'bounce' artefacts that are corrected for using empirical functions. While technical in scope, the paper purports to also offer some scientific insight to atmospheric processes at this site, so is suitable for ACP. However, I do have a misgiving on how the differences between the two instruments is interpreted (see below),

[Figure]

and I worry that the overall quality of the data is not fully validated. I would like to see that the authors more convincingly prove that the differences are due to the size of the particles and not a technical issue associated with one or more of the instruments before this proceeds to full publication. It is particularly troubling that inconsistencies are found with a number of the comparisons (e.g. MARGA, OCEC), so these should also be resolved before strong conclusions are made concerning disagreements in the two ACSMs.

Major comments:

A core part of the analysis is the assumption that any differences between the two instruments are due to a difference in the size cuts of the aerodynamic lenses. However, there are other differences between the two instruments, most notably the CV. I also worry because I see plenty of reasons to suspect that a different technical issue may be at play. The fact that the ratio between the two instruments is so consistent would imply that the PM2.5/PM1 ratio of the particulates is effectively constant, which does not seem intuitive to me. I would also expect the fractional contribution of primary OA to the PM2.5 measurement to be much lower, given that this is generally accepted to be almost entirely submicron. In order for the conclusions to stand, it would be useful if some more validation work could be presented and allay my suspicions concerning data quality. Can any data be shown where the two ACSM instruments agree for smaller particles, e.g. when dominated by primary particles? More generally, while the r2 comparisons between various instruments are certainly impressive, it might be more informative to look at the ratio in real time and see what drives this ratio to vary, e.g. if it correlates with primary vs secondary particles. This would be more informative than simply looking at a slope and speculating.

What is extremely notable in its omission is a volume-convolved particle size distribution from the DMPS/APS data. To understand the split between PM1 and PM1-2.5, it would seem fairly logical to see if the accumulation mode straddled 1 $\mu$m point. Also, because of the breadth of instrumentation at the authors' disposal, it should be possible

to do a full size-resolved mass budget by assigning components to different modes.

The discussion is a little rambling and doesn't really highlight what the new scientific advances relative to atmospheric science are. As a case in point, section 3.5 concludes that secondary aerosol are formed regionally and primaries are formed locally, which I would consider pointing out the obvious and as such, I would feel the need to question the point of presenting this. I'm not saying that the analysis should be taken out, but it should be tightened up and focused on specific points that feed into the discussion because right now, it feels like a lot of analysis just being done for the sake of it.

Minor comments:

Line 71: Chloride has also been shown to be an important contributor

Line 140: Should specify the ACSMs are the Q-ACSM type (as opposed to TOF-ACSM)

Line 157: Jayne et al. (2000) is not an appropriate reference describing the bounce effect, as it was not understood fully at the time.

Line 178: More detail is needed concerning the online OCEC measurement. Specifically the model number, the sampling duration, whether a denuder was used, which specific thermal ramp was used (specifically whether the 'abbreviated' NIOSH method was employed) and how it was calibrated. Also, from the perspective of validating the integrity of the split points, it would be useful to know the consistency between the optical and thermal EC values.

Line 182: The manufacturers and model numbers of the DMAs should be given, or if they were custom-made, the geometry employed.

Line 245: It is also likely that the size cut of the two inlets isn't identical, so this may contribute to a discrepancy.

Line 260: What did the ion balance of the MARGA data look like? If this didn't balance, this would indicate an issue with this instrument.

Line 285: While some of the reasons offered for the discrepancy between ACSM and OCEC are plausible, the fact that the correlation is so good would imply that a systematic issue is responsible. How confident are the authors that the instrument is determining the split points correctly?

Line 399: In order to show that the relationship with RH is causal, you must rule out confounding factors like changing source regions being responsible (these would have an effect on both humidity and precursor emissions). Otherwise, a caveat should be added.

Line 423: Information on the relative uses of gasoline and diesel in Nanjing should be discussed; the former is mainly responsible for the CO, but the latter is responsible for POA. While they will still correlate as an area source, the ratio will vary according to the emissions fleet (Allan et al., 2010).

---

## Referee Comment (RC2) · Anonymous Referee #2 · 11 Jun 2017

Comments on "Field characterization of the PM$_{2.5}$ Aerosol Chemical Speciation Monitor: insights into the composition, sources and processes of fine particles in Eastern China"

This Paper described the first comparison results of ACMS equipped with newly developed PM2.5 lens +capture vaporizer with other multiple on-line instruments, including a traditional PM1 ACSM (with standard vaporizer), TDMPS (for particle size distribution), On-line EC/OC, MARGA (for inorganic species), TOEM (total PM2.5 mass), BAM etc. Apparently, this is a sufficient and valuable dataset to investigate the performance of PM2.5-ACSM. Good linear correlations have been shown between the non-refractory species detected in PM2.5-ACSM with other measurements, suggesting a full detection of PM2.5 masses in this type of ACSM. Then the authors discussed secondary inorganic aerosol formation, POA and SOA, the aqueous/photochemical reactions, two case studies and the geography origins of those aerosols. Overall, I recommend this manuscript to be published in ACP. However, a major revision is suggested here based on the reason addressed below.

The authors tried to combine multiple topics into one paper, which is very distracting. I do not know the topic of this paper is to evaluate the PM2.5-ACSM or to investigate the aerosol formation. The analysis in the Section 3.3-3.5 is quite shallow. Exclusive similar results and analysis on aerosol composition and sources have been published in China before (Li et al., 2017 and references therefor in). I did not see any new finding in the analysis reported in this paper. I suggest the authors cut and combine these parts and focus on more interesting points. For the comparison part, the authors should pay more attention to the details for validating their results, since this is the main selling point based on the abstract.

**Major comments:**

 (1) Line 248-267: I found the comparison ratios of PM1-ACSM vs PM2.5-ACSM for different species are quite different in Fig. 2. i.e. OA: 0.5~; sulfate ~0.35; nitrate ~0.72; Ammonium ~0.46; Chloride: ~1.  Any comment on this? To explain this difference is quite important for understanding the PM2.5-ACSM instrument. Does this indicate the aerosols are externally mixed? However, when I looked at the time series of mass concentration in Fig.3, nitrate correlate quite well with sulfate and ammonium, implying the aerosols might be internally mixed.

In line 397-398, the authors also stated the SOA and SIA are internally mixed. If the SOA and SIA are internally mixed, then why the ratios of PM1 vs PM2.5 are varied with different inorganic and organic aerosol species. What are the PM1 vs PM2.5 ratios of OOA and POA?

(2) Line 298-300: Since the authors have the TDMPS, how is the volume comparison between PM1, PM1-2.5 and PM2.5 based on the TDMPS data..

More importantly, This dataset, as another independent dataset, can help to confirm the comparison results between ACSM with BAM and TOEM. The authors should calculate the PM1 and PM2.5 masses based on volume conc. calculated based on TDMPS measurement and

density calculated from ACSM, which can help the quantification of total PM1-ACMS and PM2.5 ACSM in Fig. 1 and Fig. 2.

(3) The authors should quantify if the PM2.5-ACMS really detect the PM2.5 masses. How much mass was lost at smaller size ranges in the PM2.5-ACMS. Xu et al. (2016) showed a larger mass loss below 200 nm of aerodynamic particle size. The author can calculate the lost masses based on TDMPS size distribution measurement.

(4) Line 212-216: Which calculation mode of ISORROPIA-II did the authors use for this calculation? Is there any gas-phase measurement to constrain the input or evaluate the output of the model? Gas phase $NH_3$ was also reported in this paper. Has the author compare the modeled NH3 with measurement NH3 to validate their results?

(4) Line 432-437, The $NO_2$ accelerate the sulfate formation is based on the fact that the aerosol is neutral. What is the pH of aerosol in this study? The authors had run the ISORROPIA-II, thus the pH should be easily calculated (Guo et al. 2017).

(5) Line 283-284: Since the authors have calibrated the instrument with NH4NO3 particles, the authors can derive their own CO2/NO3 ratio following Pieber et al. (2016).

Line 353-355: The authors should estimate their own fCO2 interferences based on the calibration data. The fCO2 production from other crustal nitrate can be roughly estimated based on the relationship between CO2/NO3 from pure NH4NO3 particles and CO2/NO3 from pure NaNO3 or other particles reported in Pieber et al. (2016).

(6) Line 260-264: This is a paradox: if all the Na came from NaCl, then it will not exist as NaNO3. The author cannot assume all Na+ exist in forms of NaNO3 in the aerosols then exist as NaCl at the same time. Meanwhile, (1) the author can assume a maximum Cl mass balanced from Na, Ca, K+, Mg. To see if this calculated maximum Cl can explain the difference of Cl between AMS and Marga. (2) the author could correlate the time series of $NO_3$ difference between AMS and MARGA with that of Cl. In such a way, the authors could check if these differences come from the same source.

**Other comments:**

Line 54: Please define "high time-resolution"

Line 78-80: A paper published recently suggested the aerosol pH in Beijing is less than 5, typically close to 4, even under the highest levels of ammonia. This level of acidity suppresses potential multi-phase sulfur oxidation pathways recently suggested to explain missing sulfate sources in the region (Liu et al. 2017). The author should also consider the possibility of this point.

Line 88-89: Any evidence for this? Is there any other potential reason that could lead to this difference of source apportionment?

Line 160: Change "response factor" to be "ionization efficiency (IE)" or "sensitivity"

Line 243-245: To better address the comparison result, the comparison uncertainty, propagated from the measurement uncertainty of each instrument should be fully addressed, which can give a better understanding of how good of the comparison results and also will be useful references for other users.

Line 271-278: What the size cut of on-line EC/OC instrument. If it is PM2.5, then the ratio between OM from PM1-ACSM vs OC from PM2.5 EC/OC is meaningless, which should not be considered at all.

Line 276-278: The authors did not show any evidence to support this statement.

Line 344: What is the f60 and f73 from the PM2.5-ACSM-CV compared with these from the PM1-ACSM-SV.

Line 345: Specify the m/z 55/57 ratio value here.

Line 386-389: Have the authors considered new particle formation process could be a potential reason for the higher NH4/NH3+NH4 ratios at smaller size ranges.

Line 380: Delete "e"

Line 393-395: What is the "different roles"? Please specify.

Line 398-399: What kind of SOA enhancement? I only saw an enhanced SOA at lower LWC concentration (<50 ug/m3) in Fig 8a.

Line 419-423: The authors stated cooking emission included in the POA factor and POA showed clearly noon and night peaks for cooking emissions. Thus, the POA/CO ratios should not be dominated by urban traffic emissions. In contrast, I think the POA vs CO regression ratio reported here should be larger than POA vs CO ratio from urban traffic emissions, based on Hayes et al. 2013.

What is the regression method for the POA/CO calculation. Could the authors show the scatter plots. Orthogonal distance regression should be used. Is the intercept fitted to be zero or not?

**References:**

Liu, M., Song, Y., Zhou, T., Xu, Z., Yan, C., Zheng, M., Wu, Z., Hu, M., Wu, Y., and Zhu, T.: Fine particle pH during severe haze episodes in northern China, Geophys Res Lett, n/a-n/a, 10.1002/2017gl073210, 2017.

Hayes, P. L., Ortega, A. M., Cubison, M. J., Froyd, K. D., Zhao, Y., Cliff, S. S., Hu, W. W., Toohey, D. W., Flynn, J. H., Lefer, B. L., Grossberg, N., Alvarez, S., Rappenglück, B., Taylor, J. W., Allan, J. D., Holloway, J. S., Gilman, J. B., Kuster, W. C., de Gouw, J. A., Massoli, P., Zhang, X., Liu, J., Weber, R. J., Corrigan, A. L., Russell, L. M., Isaacman, G., Worton, D. R., Kreisberg, N. M., Goldstein, A. H., Thalman, R., Waxman, E. M., Volkamer, R., Lin, Y. H.,

Surratt, J. D., Kleindienst, T. E., Offenberg, J. H., Dusanter, S., Griffith, S., Stevens, P. S., Brioude, J., Angevine, W. M., and Jimenez, J. L.: Organic aerosol composition and sources in Pasadena, California, during the 2010 CalNex campaign, Journal of Geophysical Research: Atmospheres, 118, 9233-9257, 10.1002/jgrd.50530, 2013.

Guo, H., Liu, J., Froyd, K. D., Roberts, J. M., Veres, P. R., Hayes, P. L., Jimenez, J. L., Nenes, A., and Weber, R. J.: Fine particle pH and gas–particle phase partitioning of inorganic species in Pasadena, California, during the 2010 CalNex campaign, Atmos. Chem. Phys., 17, 5703-5719, 10.5194/acp-17-5703-2017, 2017.

Li, Y. J., Sun, Y., Zhang, Q., Li, X., Li, M., Zhou, Z., and Chan, C. K.: Real-time chemical characterization of atmospheric particulate matter in China: A review, Atmos Environ, 158, 270-304, https://doi.org/10.1016/j.atmosenv.2017.02.027, 2017.

---

## Author Comment (AC1) · 27 Aug 2017

Dear Editor,

Thank you very much for your and reviewers' thoughtful and constructive comments on our manuscript! We have revised the manuscript accordingly. The detailed responses to the comments are given below, point by point, in blue, and the revised manuscript tracked changes in red.

**Response to Reviewer #1**

This data shows a comparison of ACSM data in Nanjing between a traditional $PM_1$ instrument and a new $PM_{2.5}$ instrument equipped with a capture vaporiser (CV), along with other instruments. These two modifications to the instrument design have been heavily anticipated for the purposes of comparability with other $PM_{2.5}$ measurements and to get away from the 'bounce' artefacts that are corrected for using empirical functions. While technical in scope, the paper purports to also offer some scientific insight to atmospheric processes at this site, so is suitable for ACP.

We thank the reviewer for his/her constructive comments on this manuscript. As detailed below, we took each of these comments into account to revise the proposed article.

However, I do have a misgiving on how the differences between the two instruments is interpreted (see below), and I worry that the overall quality of the data is not fully validated. I would like to see that the authors more convincingly prove that the differences are due to the size of the particles and not a technical issue associated with one or more of the instruments before this proceeds to full publication. It is particularly troubling that inconsistencies are found with a number of the comparisons (e.g. MARGA, OCEC), so these should also be resolved before strong conclusions are made concerning disagreements in the two ACSMs.

Thanks to reviewer comments, and as presented hereafter, we have reinforced discussions about data validation and data quality. We also now provide some new figures in both the main text and supplementary and related discussions to investigate the differences between the two Q-ACSMs with MARGA and OC/EC analyzers.

Major comments:

A core part of the analysis is the assumption that any differences between the two instruments are due to a difference in the size cuts of the aerodynamic lenses. However, there are other differences between the two instruments, most notably the CV. I also worry because I see plenty of reasons to suspect that a different technical issue may be at play. The fact that the ratio between the two instruments is so consistent would imply that the $PM_{2.5}/PM_1$ ratio of the particulates is effectively constant, which does not seem intuitive to me. I would also expect the fractional contribution of primary OA to the $PM_{2.5}$ measurement to be much lower, given that this is generally accepted to be almost entirely submicron. In order for the conclusions to stand, it would be

useful if some more validation work could be presented and allay my suspicions concerning data quality. Can any data be shown where the two ACSM instruments agree for smaller particles, e.g. when dominated by primary particles? More generally, while the $r^2$ comparisons between various instruments are certainly impressive, it might be more informative to look at the ratio in real time and see what drives this ratio to vary, e.g. if it correlates with primary vs secondary particles. This would be more informative than simply looking at a slope and speculating.

We thank the referee's comments. New Figure 4 is now showing the relationship between the species-dependent ratios of [PM$_1$-Q-ACSM] / [PM$_{2.5}$-Q-ACSM] and the ratio of [PM$_{2.5}$ SOA] / [total PM$_{2.5}$ OA]. As presented in Figure 4, each PM$_1$-to-PM$_{2.5}$ species ratio (as measured by Q-ACSMs) is decreasing with the increase of SOA relative contribution. These results are consistent with an overall smaller size of primary particles compared to secondary aerosols, and suggest the absence of any systematic sampling bias between both Q-ACSMs. We have also added the following discussion about this point:

"Figure 4 shows the relationship between the PM$_1$/PM$_{2.5}$ ratios of aerosol species from Q-ACSM measurements and the ratio of SOA to OA in PM$_{2.5}$. It can be seen that the ratios of all aerosol species generally decrease as the increase of SOA/OA in PM$_{2.5}$. Given that primary particles are more abundant than SOA in smaller size ranges, our results suggest that the PM$_{2.5}$ CV and PM$_1$ SV Q-ACSMs show a better agreement for measuring smaller particles while larger particles have higher probability to bounce off the SV surface compared with the CV (Xu et al., 2017a)."

[Figure]

**Figure 4.** Relationship between the PM$_1$/PM$_{2.5}$ ratios of aerosol species from Q-ACSM measurements and the ratio of SOA to OA in PM$_{2.5}$. The error bars refer to standard deviation. The dots in grey are the raw data points corresponding to Figure 2. The mean (solid squares), 25th and 75th percentiles (lower and upper bands) are also shown.

What is extremely notable in its omission is a volume-convolved particle size distribution from the DMPS/APS data. To understand the split between $PM_1$ and $PM_{1-2.5}$, it would seem fairly logical to see if the accumulation mode straddled 1 µm point. Also, because of the breadth of instrumentation at the authors' disposal, it should be possible to do a full size-resolved mass budget by assigning components to different modes.

We have performed the volume size distribution analysis as shown in Figure R1. We now make use of the volume-convolved mass calculation from the data of TDMPS and/or APS. This was introduced within new Figure 1, and the related discussion reads as:

"As shown in Fig. 1, the mass concentrations of $PM_1$ and $PM_{2.5}$ measured by Q-ACSMs agree well with those measured by collocated instruments (i.e., the total PM mass analyzers, including TEOM-FDMS and BAM-1020) and those estimated from size-resolved particle number concentrations (TDMPS and APS) and the composition dependent particle density (Fig. S9). On average, the total dry mass of $PM_1$ and $PM_{2.5}$ Q-ACSM reports 89 % and 93 % of the volume-dependent mass, respectively (Fig. S10). As reported in Xu et al. (2017a), the $PM_{2.5}$ lens system might have a considerable loss for particles below 200 nm due to the lens transmission efficiency (on average 45 %), which can partly explain the differences between Q-ACSM and TDMPS (Fig. S10d)."

[Figure]

**Figure R1.** Aerosol particle number (a) and volume (b) size distributions (3 nm to 10 mm).

[Figure]

**Figure 1.** Comparisons between the total particle mass concentrations measured by the $PM_1$ and $PM_{2.5}$ ACSMs, a $PM_{2.5}$ TEOM-FDMS and two MET ONE BAM 1020 (for $PM_1$ and $PM_{2.5}$, respectively), as well as volume-convolved mass calculated from TDMPS and APS, i.e. $PM_1$ (~13-1000 nm), $PM_{1-2.5}$ (~1000-2500 nm), and $PM_{2.5}$ (~13-2500 nm), and particle density estimated from ACSM species. Note that NR-$PM_1$ and NR-$PM_{2.5}$ are the mass loadings of the sum of organics, nitrate, sulfate, nitrate, ammonium, and chloride from $PM_1$ and $PM_{2.5}$ ACSM, respectively.

As suggested, we also modified the supplementary information (S10), such as:

[Figure]

**Figure S10.** Correlations between $PM_1$-ACSM, $PM_{2.5}$-ACSM, $PM_1$-BAM, $PM_{2.5}$-BAM and volume-dependent mass (TDMPS and APS) with the particle density being calculated from the chemical species of $PM_1$-ACSM and $PM_{2.5}$-ACSM, respectively. On average, the $PM_1$ and $PM_{2.5}$ Q-ACSM total dry mass reports respectively 89 % and 93 % of the $PM_1$ and $PM_{2.5}$ volume-dependent mass concentrations. As reported by Xu et al. (2017a), the $PM_{2.5}$ lens system showed a significant particle loss at below around 200 nm, with a lower transmission efficiency of 45 % on average. Considering this, we estimated that the lost of small particles at size ~13 – 201 nm might account for around 3 % of the total volume-dependent $PM_{2.5}$ mass (Fig. S10d).

The discussion is a little rambling and doesn't really highlight what the new scientific advances relative to atmospheric science are. As a case in point, section 3.5 concludes that secondary aerosol are formed regionally and primaries are formed locally, which I would consider pointing out the obvious and as such, I would feel the need to question the point of presenting this. I'm not saying that the analysis should be taken out, but it should be tightened up and focused on specific points that feed into the discussion because right now, it feels like a lot of analysis just being done for the sake of it.

Thanks for the reviewer's comments. We have re-organized the manuscript following all the comments and considering the following points:

- The deployment of AMS-type instruments for highly time-resolved chemical evolution processes of NR-$PM_1$ species have been rapidly growing over the world, which has greatly improved our understanding of the key atmospheric processes of aerosol particles during the last ten years. Limited by the aerodynamic lens, previous AMS and Q-ACSM only measure aerosol species in $PM_1$. This is reasonable for the studies in the US and Europe where $PM_1$ accounts for a large fraction (typically > 70 %) of $PM_{2.5}$. However, a substantial fraction of aerosol particles in 1–2.5 μm ($PM_{1–2.5}$) is frequently observed in China, and the contribution can be more than 50 % during severe haze events. The source apportionment results of $PM_1$ might have differences from $PM_{2.5}$ by missing such a large fraction of aerosol particles in $PM_{1-2.5}$. Therefore, the instruments which can measure $PM_{2.5}$ composition in real-time are urgently needed in China for a better understanding of variations, sources, and formation mechanisms of $PM_{2.5}$.

- In this study, we firstly evaluate the performance of the new $PM_{2.5}$ Q-ACSM that is equipped with a $PM_{2.5}$ aerodynamic lens and a capture vaporizer (CV) system. As indicated by the inter-comparisons between the $PM_{2.5}$ Q-ACSM and collocated instruments (including the $PM_{2.5}$ MARGA, $PM_1$ Q-ACSM, TDMPS, APS, and total mass analyzers), we found that the two Q-ACSMs illustrated similar temporal variations between $PM_1$ and $PM_{2.5}$ for all non-refractory species, yet substantial mass fractions of aerosol species were observed in the size range of 1–2.5 μm.

- Meanwhile, we found insignificant influence of the data discrepancy between the $PM_{2.5}$ Q-ACSM and MARGA on fine aerosol pH prediction by using ISORROPIA-II. The fine aerosol pH in this study

ranged from 2.4 to 4.3indicating acidic aerosol particles. This result suggests that the recently proposed aqueous oxidation of $SO_2$ by $NO_2$ under neutral conditions (Wang et al., 2016) during severe haze pollution is not important for sulfate formation during several haze episodes in this study.

- In addition, positive matrix factorization of organic aerosol showed similar temporal variations in both primary and secondary OA between $PM_1$ and $PM_{2.5}$ although the mass spectra were slightly different due to more thermal decomposition on the capture vaporizer of $PM_{2.5}$-Q-ACSM. The impacts of aqueous and photochemical processes on the formation of SOA are also discussed.

In this respect, the "results and discussion" section is now divided as follows:

3.1 Inter-comparisons

3.2 Size-segregated investigation of NR-$PM_{2.5}$ components

3.2.1 Secondary inorganic aerosols

3.2.2 POA and SOA

3.3 Specific episodes analysis (Ep2 and Ep5)

**Minor comments:**

Line 71: Chloride has also been shown to be an important contributor

Chloride is typically important in biomass burning events, but contributes a small fraction of the total $PM_1$ (< 5 %) for most studies (Li et al., 2017). Here we refer to secondary inorganic aerosols, while chloride is dominantly from primary emissions, e.g., biomass burning, coal combustion and sea salts. We have modified the related sentence. It now reads as:

"Secondary organic aerosols (SOA) and secondary inorganic aerosols (e.g., sulfate, nitrate, and ammonium)…"

Line 140: Should specify the ACSMs are the Q-ACSM type (as opposed to TOF-ACSM)

Specified. The ACSM has been defined as Q-ACSM throughout the manuscript.

Line 157: Jayne et al. (2000) is not an appropriate reference describing the bounce effect, as it was not understood fully at the time.

This reference has been removed.

Line 178: More detail is needed concerning the online OCEC measurement. Specifically the model number, the sampling duration, whether a denuder was used, which specific thermal ramp was used (specifically whether the 'abbreviated' NIOSH method was employed) and how it was calibrated. Also, from the perspective of validating the integrity of the split points, it would be useful to know the consistency between the optical and thermal EC values. + Line 285: While some of the reasons offered for the discrepancy between ACSM and OCEC are plausible, the fact that the correlation is so good would imply that a systematic issue is responsible. How confident are the authors that the instrument is determining the split points correctly?

Thank you for this comment, which is also quite helpful. The mass concentrations of OC and EC in $PM_{2.5}$ were measured on a 1hr-basis using a Sunset Lab. Semi-Continuous OCEC Analyzer (Model-4) implemented with the standard 'abbreviated' NIOSH 5040 thermal protocol (as detailed in Table S1). A denuder was placed in the sampling line to remove volatile organic compounds and avoid positive sampling artefacts.

**Table S1.** Thermal protocol used in this study within the Sunset Lab. Semi-Continuous OC/EC Analyzer

| Gas | Hold time (s) | Temperature (°C) |
|---|---|---|
| He | 10 | 1 |
| He | 95 | 600 |
| He | 95 | 840 |
| He | 30 | Oven off |
| He | 5 | 550 |
| He/$O_2$ | 10 | 550 |
| He/$O_2$ | 25 | 550 |
| He/$O_2$ | 45 | 650 |
| He/$O_2$ | 115 | 870 |

The stock sucrose solution was used in this study for non-dispersive infrared (NDIR) calibration.

Figure R2 presents the time series of the thermal and optical EC and OC as furnished by the Sunset Field Inst. Although both datasets show very good correlations, thermal EC are about 3 times higher than optical EC measurements, while thermal OC is a bit lower than optical OC estimates. Such a phenomenon could be related to poor calibration of the oven temperature (e.g., Panteliadis et al., 2015), which couldn't be checked before nor after the campaign.

However, differences between thermal and optical EC measurements are relatively small compared to total $PM_{2.5}$ concentrations (Fig. R2c), so that using thermal or optical EC data sets has no significant influence on results retrieved from $PM_{2.5}$ mass closures exercises.

On the other hand, as shown in new Fig. 3, the ratio between $PM_{2.5}$ OA ACSM data and optical OC estimates is only slightly smaller than the one obtained for thermal OC measurements (2.9 vs. 3.5), which cannot fully elucidate discrepancies observed between the two instruments.

[Figure]

**Figure R2.** (a-b) time series of the thermal and optical OC/EC measured by OC/EC analyzer, and (c) the relationship between the EC difference between Thermal and Optical.

[Figure]

**Figure 3.** Scatter plots with the linear regression parameters and the 1:1 line (dash line) shown for the comparisons. Note that the term of "PM$_{2.5}$" in the plot of Fig. 3a means that the sum mass concentration of PM$_{2.5}$-ACSM species (organics, nitrate, sulfate, ammonium, and chloride), Sunset EC, and MARGA species (K$^+$, Na$^+$, Mg$^{2+}$, and Ca$^{2+}$). Also note that differences observed between "thermal" and "optical" PM$_{2.5}$ OC measurements (Fig. 3b) might be related to poor calibration of the oven temperature probe (e.g., Panteliadis et al., 2015), which couldn't be checked before nor after the campaign.

Finally, as reported from the ACTRIS Q-ACSM inter-comparison works, it should also be noted that instrument artifacts may significantly affect the variability in $f_{44}$ measured by different Q-ACSMs (Crenn et

al., 2015). Although such artifact is not clear from the present work, future studies should investigate it from both laboratory and field experiments onto PM$_{2.5}$ Q-ACSM equipped with CV.

Line 182: The manufacturers and model numbers of the DMAs should be given, or if they were custom-made, the geometry employed.

The model number of the DMAs has been given.

Line 245: It is also likely that the size cut of the two inlets isn't identical, so this may contribute to a discrepancy.

We thank the reviewer's comments. It could be one of the reasons. We have modified the main text. It now reads in the main text as:

"This slight underestimation of the total PM$_{2.5}$ mass might be primarily due to discrepancies between the different inlet cut-offs, measurement uncertainties of the different instruments as further discussed below, the un-identified mineral dust and sea salt components."

Line 260: What did the ion balance of the MARGA data look like? If this didn't balance, this would indicate an issue with this instrument.

As shown in Fig.4, the equivalent ratios of anions to cations ranged from 0.98 to 1.05 with and without crustal species (K$^+$, Na$^+$, Ca$^{2+}$, and Mg$^{2+}$), suggesting an excellent ion balance of the MARGA measurements. This supports the data quality of the MARGA measurements.

[Figure]

**Figure S8.** Ion balance of the water-soluble ions measured by the PM$_{2.5}$ MARGA. Note that: anion equivalents = [NH$_4^+$/18] + [Na$^+$/23] + [K$^+$/39] + 2 × [Mg$^{2+}$/24] + 2 × [Ca$^{2+}$/40], and cation equivalents = 2 × [SO$_4^{2-}$/96] + [NO$_3^-$/62] + [Cl$^-$/35.5], in which chemical ions are in the unit of μg/m$^3$.

Line 399: In order to show that the relationship with RH is causal, you must rule out confounding factors like changing source regions being responsible (these would have an effect on both humidity and precursor emissions). Otherwise, a caveat should be added.

That's another good suggestion. We have performed wind-dependent analysis (a new Fig. 10) for RH, aerosol chemical species, and gas-phase precursors to investigate their regional sources. Figure 10 presents results obtained from the nonparametric wind regression analysis performed following the procedures described in Petit et al. (2017). High RH levels (> 80 %) and ALWC (> 30 µg.m$^{-3}$ for PM$_1$ and 50 µg.m$^{-3}$ for PM$_{2.5}$) are mainly associated with Northwestern air masses, the later ones being loaded with relatively high amounts of secondary aerosols (SOA, as well as nitrate and sulfate) but low amount of gas-phase precursors (e.g., O$_3$, SO$_2$, NO$_2$, and NH$_3$). These results suggest the predominance of aqueous phase chemistry in SOA formation from the Northwestern sector.

[Figure]

**Figure 10.** Analysis on relative humidity (RH), and gas-phase species (SO$_2$, NO$_2$, O$_3$, NH$_3$, and CO) and PM$_1$ and PM$_{2.5}$ ALWC, SOA, and [SO$_4$ + NO$_3$], respectively. Radius and angle of each plot refers to wind speed and wind direction.

Line 423: Information on the relative uses of gasoline and diesel in Nanjing should be discussed; the former is mainly responsible for the CO, but the latter is responsible for POA. While they will still correlate as an area source, the ratio will vary according to the emissions fleet (Allan et al., 2010).

Considering all comments from both reviewer and due to the new outline of this manuscript, the discussion about the relationship between OA to CO has been removed.

**Response to Reviewer #2**

This Paper described the first comparison results of ACMS equipped with newly developed PM2.5 lens +capture vaporizer with other multiple on-line instruments, including a traditional PM1 ACSM (with standard vaporizer), TDMPS (for particle size distribution), On-line EC/OC, MARGA (for inorganic species), TOEM (total $PM_{2.5}$ mass), BAM etc. Apparently, this is a sufficient and valuable dataset to investigate the performance of $PM_{2.5}$-ACSM. Good linear correlations have been shown between the non-refractory species detected in $PM_{2.5}$-ACSM with other measurements, suggesting a full detection of $PM_{2.5}$ masses in this type of ACSM. Then the authors discussed secondary inorganic aerosol formation, POA and SOA, the aqueous/photochemical reactions, two case studies and the geography origins of those aerosols. Overall, I recommend this manuscript to be published in ACP. However, a major revision is suggested here based on the reason addressed below.

We thank the reviewer for his/her helpful comments.

The authors tried to combine multiple topics into one paper, which is very distracting. I do not know the topic of this paper is to evaluate the $PM_{2.5}$-ACSM or to investigate the aerosol formation. The analysis in the Section 3.3-3.5 is quite shallow. Exclusive similar results and analysis on aerosol composition and sources have been published in China before (Li et al., 2017 and references therefor in). I did not see any new finding in the analysis reported in this paper. I suggest the authors cut and combine these parts and focus on more interesting points. For the comparison part, the authors should pay more attention to the details for validating their results, since this is the main selling point based on the abstract.

We thank the reviewer for very helpful comments and suggestions. We have re-organized the manuscript following all the comments and considering the following points:

- The deployment of AMS-type instruments for highly time-resolved chemical evolution processes of NR-$PM_1$ species have been rapidly growing over the world, which has greatly improved our understanding of the key atmospheric processes of aerosol particles during the last ten years. Limited by the aerodynamic lens, previous AMS and Q-ACSM only measure aerosol species in $PM_1$. This is reasonable for the studies in the US and Europe where $PM_1$ accounts for a large fraction (typically > 70 %) of $PM_{2.5}$. However, a substantial fraction of aerosol particles in 1–2.5 μm ($PM_{1–2.5}$) is frequently observed in China, and the contribution can be more than 50 % during severe haze events. The source apportionment results of $PM_1$ might have differences from $PM_{2.5}$ by missing such a large fraction of aerosol particles in $PM_{1-2.5}$. Therefore, the instruments which can measure $PM_{2.5}$ composition in real-time are urgently needed in China for a better understanding of variations, sources, and formation mechanisms of $PM_{2.5}$.

- In this study, we firstly evaluate the performance of the new $PM_{2.5}$ Q-ACSM that is equipped with a $PM_{2.5}$ aerodynamic lens and a capture vaporizer (CV) system. As indicated by the inter-comparisons between the $PM_{2.5}$ Q-ACSM and collocated instruments (including the $PM_{2.5}$ MARGA, $PM_1$ Q-

ACSM, TDMPS, APS, and total mass analyzers), we found that the two Q-ACSMs illustrated similar temporal variations between $PM_1$ and $PM_{2.5}$ for all non-refractory species, yet substantial mass fractions of aerosol species were observed in the size range of 1–2.5 µm.

- Meanwhile, we found insignificant influence of the data discrepancy between the $PM_{2.5}$ Q-ACSM and MARGA on fine aerosol pH prediction by using ISORROPIA-II. The fine aerosol pH in this study ranged from 2.4 to 4.3indicating acidic aerosol particles. This result suggests that the recently proposed aqueous oxidation of $SO_2$ by $NO_2$ under neutral conditions (Wang et al., 2016) during severe haze pollution is not important for sulfate formation during several haze episodes in this study.

- In addition, positive matrix factorization of organic aerosol showed similar temporal variations in both primary and secondary OA between $PM_1$ and $PM_{2.5}$ although the mass spectra were slightly different due to more thermal decomposition on the capture vaporizer of $PM_{2.5}$-Q-ACSM. The impacts of aqueous and photochemical processes on the formation of SOA are also discussed.

In this respect, the "results and discussion" section is now divided as follows:

3.1 Inter-comparisons

3.2 Size-segregated investigation of NR-$PM_{2.5}$ components

3.2.1 Secondary inorganic aerosols

3.2.2 POA and SOA

3.3 Specific episodes analysis (Ep2 and Ep5)

Major comments:

(1) Line 248-267: I found the comparison ratios of $PM_1$-ACSM vs $PM_{2.5}$-ACSM for different species are quite different in Fig. 2. i.e. OA: 0.5~; sulfate ~0.35; nitrate ~0.72; Ammonium ~0.46; Chloride: ~1. Any comment on this? To explain this difference is quite important for understanding the $PM_{2.5}$-ACSM instrument. Does this indicate the aerosols are externally mixed? However, when I looked at the time series of mass concentration in Fig. 3, nitrate correlate quite well with sulfate and ammonium, implying the aerosols might be internally mixed.

As tentatively discussed in section 3.1, the different $PM_1$/$PM_{2.5}$ ratios between OA, sulfate, nitrate, ammonium, and chloride are mainly due to the size-dependent chemical species. For example, the mass-weighted size distributions of organics, sulfate and nitrate can be very different, leading to different ratios of $PM_1$ and $PM_{2.5}$. Unfortunately, the size-resolved composition data was not available in this study. Another reason is likely due to the different lens transmission efficiencies for small particles (e.g., < 200 nm). Xu et al. (2017a) showed that the $PM_{2.5}$ aerodynamic lens can have a substantial loss (45% on average) of small particles below 200 nm, while the $PM_1$ aerodynamic lens has an approximately 100 % transmission efficiency for particles between 60

– 200 nm. It is quite difficult to tell the mixing states of aerosol particles according to the different ratios, but likely nitrate, sulfate and ammonium were internally mixed based on their correlations. Future studies by measuring the size-resolved bulk composition and single particle aerosol composition are needed to address these differences. Also according to the comment (7) of "other comments", we modified the revised manuscript as follows:

"As shown in Fig. S13a, we estimated that about 83 % of the difference between chloride $PM_{2.5}$-Q-ACSM and MARGA measurements could be explain by a maximum estimate of refractory chloride calculated using the ion mass balance with $Na^+$, $Ca^{2+}$, $K^+$, and $Mg^{2+}$. In addition, this estimated maximum refractory-chloride concentrations also show a positive relationship ($r^2 = 0.36$) with the difference between nitrate loadings obtained from the $PM_{2.5}$-Q-ACSM and MARGA (Fig. S13b). The presence of refractory-chloride (-nitrate) may then explain a large fraction of the discrepancies observed for these species between both $PM_{2.5}$ chemical analyzers."

In line 397-398, the authors also stated the SOA and SIA are internally mixed. If the SOA and SIA are internally mixed, then why the ratios of $PM_1$ vs $PM_{2.5}$ are varied with different inorganic and organic aerosol species.

We further analyzed the ratio of $PM_{2.5}$ SOA to inorganic species as measured using the MARGA instrument. As shown in new Fig. 9b, the SOA to [sulfate + nitrate] ratios are 0.57 and 0.39 at low RH (RH < 40 %) and high RH (RH > 80 %), respectively, i.e., in very good agreement with those obtained from the $PM_1$ Q-ACSM data (0.58 and 0.41, respectively). Therefore, the different ratios of SOA to [sulfate + nitrate] in $PM_1$ and $PM_{2.5}$ ACSM data sets could be mainly due to the underestimated nitrate measured by the $PM_{2.5}$ Q-ACSM compared to the MARGA.

[Figure]

**Figure 9b.** SOA versus $SO_4^{2-}$ + $NO_3^-$. The regression slopes at different RH levels, i.e., high RH (RH > 80 %) and low RH (RH < 40 %) and in different size ($PM_1$ and $PM_{2.5}$) are also shown.

What are the PM$_1$ vs PM$_{2.5}$ ratios of OOA and POA?

As shown in Fig. R4, the PM$_1$ POA (SOA) to PM$_{2.5}$ POA (SOA) is 0.62 (0.42).

[Figure]

**Figure R4.** The PMF-resolved SOA and POA relationship between PM$_1$ and PM$_{2.5}$.

(2) Line 298-300: Since the authors have the TDMPS, how is the volume comparison between PM$_1$, PM$_{1-2.5}$ and PM$_{2.5}$ based on the TDMPS data. More importantly, this dataset, as another independent dataset, can help to confirm the comparison results between ACSM with BAM and TOEM. The authors should calculate the PM$_1$ and PM$_{2.5}$ masses based on volume conc. calculated based on TDMPS measurement and density calculated from ACSM, which can help the quantification of total PM$_1$-ACMS and PM$_{2.5}$ ACSM in Fig. 1 and Fig. 2.

We calculated volume-convolved mass of particles from TDMPS and APS to support the data validation, and modified the manuscript as follows:

"As shown in Fig. 1, the mass concentrations of PM$_1$ and PM$_{2.5}$ measured by Q-ACSMs agree well with those measured by collocated instruments (i.e., the total PM mass analyzers, including TEOM-FDMS and BAM-1020) and those estimated from size-resolved particle number concentrations (TDMPS and APS) and the composition dependent particle density (Fig. S9). On average, the total dry mass of PM$_1$ and PM$_{2.5}$ Q-ACSM reports 89 % and 93 % of the volume-dependent mass, respectively (Fig. S10). As reported in Xu et al. (2017a), the PM$_{2.5}$ lens system might have a considerable loss for particles below 200 nm due to the lens transmission efficiency (on average 45 %), which can partly explain the differences between Q-ACSM and TDMPS (Fig. S10d)."

[Figure]

**Figure 1.** Comparisons between the total particle mass concentrations measured by the PM$_1$ and PM$_{2.5}$ ACSMs, a PM$_{2.5}$ TEOM-FDMS and two MET ONE BAM 1020 (for PM$_1$ and PM$_{2.5}$, respectively), as well as volume-convolved mass calculated from TDMPS and APS, i.e. PM$_1$ (~13-1000 nm), PM$_{1-2.5}$ (~1000-2500 nm), and PM$_{2.5}$ (~13-2500 nm), and particle density calculated by the ACSM species. Note that NR-PM$_1$ and NR-PM$_{2.5}$ are the mass loadings of the sum of organic, nitrate, sulfate, nitrate, ammonium, and chloride from PM$_1$ and PM$_{2.5}$ ACSM, respectively.

(3) The authors should quantify if the PM$_{2.5}$-ACMS really detect the PM$_{2.5}$ masses. How much mass was lost at smaller size ranges in the PM$_{2.5}$-ACSM. Xu et al. (2016) showed a larger mass loss below 200 nm of aerodynamic particle size. The author can calculate the lost masses based on TDMPS size distribution measurement.

As suggested here, we estimated that the lost small particles at size ~13 – 201 nm might account for around 3 % of the total volume-dependent PM$_{2.5}$ mass and modified the revised supplementary information as:

[Figure]

**Figure S10:** Correction between PM₁-Q-ACSM, PM₂.₅-ACSM, PM₁-BAM, PM₂.₅-BAM and volume-dependent mass (TDMPS and APS) with the particle density calculated from the chemical species of PM₁-Q-ACSM and PM₂.₅-ACSM, respectively. On average, the PM₁ and PM₂.₅ Q-ACSM total dry mass reports respectively 89 % and 93 % of the PM₁ and PM₂.₅ volume-dependent mass concentrations. As reported by Xu et al. (2017a), the PM₂.₅ lens system showed a significant particle loss at below around 200 nm, with a lower transmission efficiency of 45 % on average. Considering this, we estimated that the lost small particles at size ~13 – 201 nm might account for around 3 % of the total volume-dependent PM₂.₅ mass (Fig. S10d).

(4) Line 212-216: Which calculation mode of ISORROPIA-II did the authors use for this calculation? Is there any gas-phase measurement to constrain the input or evaluate the output of the model? Gas phase NH₃ was also reported in this paper. Has the author compare the modeled NH₃ with measurement NH₃ to validate their results?

The "forward mode" was used in the ISORROPIA-II prediction in this study. The gas-phase NH₃ and HNO₃ measrued by the MARGA was used as inputs along with the particle-phase species meausred by the two Q-ACSMs and MARGA, respectively. As shown in new Fig. S6, the ISORROPIA-II-predicted NH₃ for the particle-species data from different instruments shows a good correlation with measured NH₃, indicative of represenative aerosol state for the modelling claculations in this study. The manuscript has then been modified as follows:

"The predicted NH₃ by ISORROPIA-II agreed well with the measured NH₃ (Fig. S5), suggesting that the aerosol phase state was representative via the thermodynamic analysis. Figure S6 presents the time series of pH for PM₁ and PM₂.₅. By excluding mineral dust and sea salt species in ISORROPIA-II, the predicted pH

was in the range of 1.23 − 4.19 (PM$_1$-Q-ACSM), 1.78 − 4.10 (PM$_{2.5}$-Q-ACSM), and 1.98 − 4.07 (PM$_{2.5}$-MARGA), with the mean values being 3.47, 3.33, and 3.42, respectively. The aerosol pH showed slight increases by 5 − 6 % except the dust-related period if crustal species were included (Fig. S7). This indicates that the aerosol pH prediction was generally consistent with the measurements from different instruments. However, the crustal species have large impacts on aerosol pH. For example, fine aerosol pH shows an evident increase from 2.8 − 3.03 to 3.7 during the dust period after the cations of Ca$^{2+}$, Mg$^{2+}$, and K$^+$ were included. Figure S8 shows excellent agreements of pH prediction with and without Na$^+$ and Cl$^-$ as the model inputs, suggesting the negligible influence of sea salts on aerosol particle acidity in this study. One reason is due to the relatively low concentrations of Na$^+$ (0 − 0.87 μg m$^{-3}$) during the campaign."

[Figure]

**Figure S5.** Correction of measured NH$_3$ and predicted NH$_3$ with inputs of PM$_1$-ACSM (without MARGA's Na$^+$, Ca$^{2+}$, K$^+$, Mg$^{2+}$), PM$_{2.5}$-ACSM (with MARGA's Na$^+$, Ca$^{2+}$, K$^+$, Mg$^{2+}$), and PM$_{2.5}$-MARGA (with Na$^+$, Ca$^{2+}$, K$^+$, Mg$^{2+}$) data, respectively, and same gas-phase HNO$_3$ and NH$_3$, ambient RH, T for all predictions.

(5) Line 432-437, The NO$_2$ accelerate the sulfate formation is based on the fact that the aerosol is neutral. What is the pH of aerosol in this study? The authors had run the ISORROPIA-II, thus the pH should be easily calculated (Guo et al. 2017).

We agree with the reviewer. We have predicted the aerosol pH values by using ISORROPIA-II (as now presented in new Figure S6). The mean aerosol pH was 3.59 ± 0.37 and 3.51 ± 0.39 from the PM$_{2.5}$ MARGA and PM$_{2.5}$-Q-ACSM data respectively. Therefore, NO$_2$ enhancement of the sulfate formation mechanism could not be a typical case in this study. In this respect, the manuscript has been modified as follows:

"The average aerosol pH was 3.59 ± 0.37 and 3.51 ± 0.39, respectively using the PM$_{2.5}$ MARGA and PM$_{2.5}$-Q-ACSM measurements as ISORROPIA-II inputs (Fig. 7), indicative of acidic aerosol particle in this study. The pH values here are consistent with that (average = 4.2) observed during haze episodes in Beijing (Liu et al., 2017). Recent studies showed that sulfate formation was more sensitive to aqueous oxidation of SO$_2$ in the

presence of high $NO_2$ and neutral conditions during the haze pollution periods in China (Cheng et al., 2016; Wang et al., 2016). However, the pH values observed in this study suggest acidic particles, indicating that the aqueous oxidation pathway of $SO_2$ by $NO_2$ to form sulfate was not favored during the haze episodes in this study."

[Figure]

**Figure S6.** Comparisons of ISORROPIA-II-predicted particle pH (a & b) and ALWC (c & d) for the data from different instruments (i.e., $PM_1$-ACSM, $PM_{2.5}$-ACSM, and $PM_{2.5}$-MARGA), respectively. $SO_4^{2-} – NO_3^-$ $– NH_4^+ – Cl^- – Na^+ – Ca^{2+} – K^+ – Mg^{2+} – HNO_3 – NH_3 – H_2O$ system and $SO_4^{2-} – NO_3^- – NH_4^+ – Cl^- – HNO_3$ $– NH_3 – H_2O$ system were used for the prediction, respectively.

(6) Line 283-284: Since the authors have calibrated the instrument with $NH_4NO_3$ particles, the authors can derive their own $CO_2/NO_3$ ratio following Pieber et al. (2016). Line 353-355: The authors should estimate their own $f_{CO2}$ interferences based on the calibration data. The $f_{CO2}$ production from other crustal nitrate can be roughly estimated based on the relationship between $CO_2/NO_3$ from pure $NH_4NO_3$ particles and $CO_2/NO_3$ from pure $NaNO_3$ or other particles reported in Pieber et al. (2016).

We have checked the calibration data from pure $NH_4NO_3$ particles (Fig. R5). The signals of *m/z* 44 was nearly two order magnitudes lower than those of $NO_3$, indicating the negligible influences of ammonium nitrate on $CO_2^+$ in this study.

[Figure]

**Figure R5.** The relationship between *m/z* 44 signal with nitrate signal from pure $NH_4NO_3$ particles measured by the $PM_{2.5}$ Q-ACSM.

(7) Line 260-264: This is a paradox: if all the Na came from NaCl, then it will not exist as $NaNO_3$. The author cannot assume all $Na^+$ exist in forms of $NaNO_3$ in the aerosols then exist as NaCl at the same time. Meanwhile, (1) the author can assume a maximum Cl mass balanced from $Na^+$, $Ca^{2+}$, $K^+$, $Mg^{2+}$. To see if this calculated maximum Cl can explain the difference of Cl between AMS and Marga. (2) the author could correlate the time series of $NO_3$ difference between AMS and MARGA with that of Cl. In such a way, the authors could check if these differences come from the same source.

Thanks for reviewer's comments and suggestion. We have made a modification according to this point which now reads in the main text as: "As shown in Fig. S13a, we estimated that about 83 % of the difference between chloride $PM_{2.5}$-Q-ACSM and MARGA measurements could be explain by a maximum estimate of refractory chloride calculated using the ion mass balance with $Na^+$, $Ca^{2+}$, $K^+$, and $Mg^{2+}$. In addition, this estimated maximum refractory-chloride concentrations also show a positive relationship ($r^2 = 0.36$) with the difference between nitrate loadings obtained from the $PM_{2.5}$-Q-ACSM and MARGA (Fig. S13b). The presence of refractory-chloride (-nitrate) may then explain a large fraction of the discrepancies observed for these species between both $PM_{2.5}$ chemical analyzers."

[Figure]

**Figure S13.** Relationship between measured nitrate and chloride difference values (i.e., $PM_{2.5}$-Marga – $PM_{2.5}$-ACSM) with estimated maximum chloride by mass banlance from $Na^+$, $Ca^{2+}$, $K^+$, and $Mg^{2+}$.

**Other comments:**

Line 54: Please define "high time-resolution"

It has been defined. It now reads as "high time-resolution (e.g., less than 1 hour)".

Line 78-80: A paper published recently suggested the aerosol pH in Beijing is less than 5, typically close to 4, even under the highest levels of ammonia. This level of acidity suppresses potential multi-phase sulfur oxidation pathways recently suggested to explain missing sulfate sources in the region (Liu et al. 2017). The author should also consider the possibility of this point.

Thanks for the reviewer's comments. As discussed above, the mean aerosol pH was of about 3.5 in this study, which is similar as the value 4 observed in Beijing (Liu et al., 2017), the value of which, however, is less than the typical aerosol pH value (~7) during China severe haze events (Wang et al., 2016). As discussed above, the aqueous oxidation of $SO_2$ by $NO_2$ chemistry pathway to form sulfate may then not be favored in Nanjing during this study along with several haze episodes. This is consistent with the finding by Liu et al. (2017) during haze episodes in Beijing.

Line 88-89: Any evidence for this? Is there any other potential reason that could lead to this difference of source apportionment?

A substantial fraction of aerosol particles in 1–2.5 μm ($PM_{1-2.5}$) is frequently observed in China, and the contribution can be more than 50 % during severe haze events (Wang et al., 2015b; Zhang et al., 2015b). The source apportionment results of $PM_1$ might have differences from $PM_{2.5}$ by missing such a large fraction of aerosol particles in $PM_{1-2.5}$. As shown in Fig. R4, the ratio of $PM_1$ (SOA) to $PM_{2.5}$ POA (SOA) is 0.62 (0.42), respectively, indicative of size-dependent source apportionment results. However, we did not find more evidence here to explain such difference, and other potential reasons might then be further investigated in future studies.

Line 160: Change "response factor" to be "ionization efficiency (IE)" or "sensitivity"

Changed it to be "sensitivity".

Line 243-245: To better address the comparison result, the comparison uncertainty, propagated from the measurement uncertainty of each instrument should be fully addressed, which can give a better understanding of how good of the comparison results and also will be useful references for other users.

Thanks for the reviewer's comments. We have added more discussion on the comparisons among those instruments, together with some suggestion above. Mode details can be found in the revised manuscript, as follows:

As shown in Fig. 1, the mass concentrations of $PM_1$ and $PM_{2.5}$ measured by Q-ACSMs agree well with those measured by collocated instruments (i.e., the total PM mass analyzers, including TEOM-FDMS and BAM-1020) and those estimated from size-resolved particle number concentrations (TDMPS and APS) and the composition dependent particle density (Fig. S9). On average, the total dry mass of $PM_1$ and $PM_{2.5}$ Q-ACSM reports 89 % and 93 % of the volume-dependent mass, respectively (Fig. S10). As reported in Xu et al. (2017a), the $PM_{2.5}$ lens system might have a considerable loss for particles below 200 nm due to the lens transmission efficiency (on average 45 %), which can partly explain the differences between Q-ACSM and TDMPS (Fig. S10d). The NR-$PM_{2.5}$ concentrations report approximately 90 % of the total $PM_{2.5}$ concentrations measured by the TEOM-FDMS and/or BAM 1020 instruments (Fig. 1a). After considering the contributions of EC and alkaline cations ($Na^+ + K^+ + Ca^{2+} + Mg^{2+}$), it can explain 92 % of the $PM_{2.5}$ mass. This slight underestimation of the total $PM_{2.5}$ mass might be primarily due to discrepancies between the different inlet cut-offs, measurement uncertainties of the different instruments, and, as further discussed below, the un-identified mineral dust and sea salt components.

Figures 2 and 3 show the inter-comparisons of the measurements by the $PM_{2.5}$-Q-ACSM with those by other co-located instruments, including $PM_1$-Q-ACSM, MARGA, and OC/EC analyzer. Overall, the $PM_{2.5}$-Q-ACSM measurements are well correlated with those measured by co-located instruments ($r^2 > 0.9$), except for chloride. SNA (= sulfate + nitrate + ammonium) measured by the $PM_{2.5}$-Q-ACSM were highly correlated with those measured by the MARGA ($r^2 = 0.92$–$0.95$). The absolute agreement between the $PM_{2.5}$-Q-ACSM and MARGA is very good for sulfate (slope = 1.02). The ammonium agreement is also quite good with the $PM_{2.5}$ Q-ACSM measuring on average 89 % of that reported by the MARGA. The average ratios of the measured $NH_4^+$ to predicted $NH_4^+$ that requires to fully neutralize $SO_4^{2-}$, $NO_3^-$, and $Cl^-$ were 1.02 and 0.95 for the $PM_{2.5}$-Q-ACSM and $PM_1$-Q-ACSM, respectively (Fig. S11), which is similar to the water-soluble ion balance results from the MARGA (Fig. S12). For nitrate, however, the $PM_{2.5}$ Q-ACSM measures about 68 % of what is reported by the MARGA. One reason might be due to the contribution of nitrate from aged sea salts and/or mineral dust (e.g., $NaNO_3$ and $Mg(NO_3)_2$) (Gibson et al., 2006), that Q-ACSM cannot detect due to the limited vaporizer temperature. The much lower ratio of chloride (0.26, Fig. 3f) between the $PM_{2.5}$-Q-ACSM and MARGA also suggests the presence of such sea salt and/or crustal particles. As shown in Fig. S13a, we estimated that about 83 % of the difference between chloride $PM_{2.5}$-Q-ACSM and MARGA measurements could be explain by a maximum estimate of refractory chloride calculated using the ion mass balance with $Na^+$, $Ca^{2+}$, $K^+$, and $Mg^{2+}$. In addition, this estimated maximum refractory-chloride concentrations also show a positive relationship ($r^2 = 0.36$) with the difference between nitrate loadings obtained from the $PM_{2.5}$-Q-ACSM and MARGA (Fig. S13b). The presence of refractory-chloride (-nitrate) may then explain a large fraction of the discrepancies observed for these species between both $PM_{2.5}$ chemical analyzers. Moreover, a recent evaluation of the AMS with a CV system also found a large difference in chloride measurements (Hu et al.,

2017), yet the reason was not completely understood. A future RIE calibration for chloride in the CV system might be helpful to evaluate these differences.

Line 271-278: What the size cut of on-line EC/OC instrument. If it is PM$_{2.5}$, then the ratio between OM from PM$_1$-ACSM vs OC from PM$_{2.5}$ EC/OC is meaningless, which should not be considered at all.

The OC/EC analyzer was indeed operated with a PM$_{2.5}$ cut-off inlet. The comparison between PM$_{2.5}$ OC data and PM$_1$ Q-ACSM OA measurements was indicated just for information. This is now removed from revised manuscript following your suggestion.

Line 276-278: The authors did not show any evidence to support this statement.

As shown in Fig. R4 and Fig. 6, we found a larger fraction of SOA in the PM$_{1-2.5}$ OA particles rather than POA, suggesting that the oxidized oxygenated OA in the PM$_{1-2.5}$ is more important than POA. This may support the statement in the manuscript. We have modified the related sentence, which now reads as:

"…and it may be expected that PM$_{1–2.5}$ organic aerosols may be more oxidized than the submicron fraction (with higher contribution of SOA, as discussed in section 3.2.2),"

[Figure]

**Figure 6.** Mass concentration (a) and fraction (b) of NR-PM$_1$ and NR-PM$_{1–2.5}$ chemical components in NR-PM$_{2.5}$ respectively during different episodes (Ep1–Ep5) marked in Figure 3 and entire study period (Total).

Line 344: What is the $f_{60}$ and $f_{73}$ from the PM$_{2.5}$-ACSM-CV compared with these from the PM$_1$-ACSM-SV.

The inter-comparison of $f_{60}$ and $f_{73}$ from the PM$_{2.5}$-ACSM-CV and the PM$_1$-ACSM-SV is now shown in Fig. R6 and a new Fig. S15 with the absolute mass. The $f_{60}$ and $f_{73}$ from the PM$_1$-ACSM-SV are obviously higher

than that from the PM$_{2.5}$-ACSM-CV. This may support that the stronger thermal decomposition of *m/z* 60 and *m/z* 73 in the PM$_{2.5}$-ACSM-CV system, as discussed in the manuscript: "Note that the mass spectra of NR-PM$_{2.5}$ shows smaller fractions of *m/z* 60 and *m/z* 73 signals, compared with those of PM$_1$ (Fig. 8a and Fig. S15),"

[Figure]

**Figure R6.** Time series of fraction of *m/z* 60 and *m/z* 73 in total OA signal.

Line 345: Specify the *m/z* 55/57 ratio value here.

The *m/z* 55 / 57 is 1.91, which has been specified in the main text.

Line 386-389: Have the authors considered new particle formation process could be a potential reason for the higher NH$_4$/NH$_3$+NH$_4$ ratios at smaller size ranges.

Thanks again for this comment. The new particle formation in such NH$_3$-rich environment could be a potential reason for the higher NH$_4$/NH$_3$+NH$_4$ ratios at smaller size ranges. For example, as shown in new Fig. 11 in the revised manuscript, the aerosol pH shows an evident peak (pH = ~4) during the new particle formation, while ALWC is very low (2.4 μg m$^{-3}$). This may suggest that heterogeneous reaction might be involved into the new particle formation process under such NH$_3$-rich environments.

The manuscript has been revised accordingly, as follows:

"Interestingly, the aerosol pH shows an evident peak (pH = ~4) during the new particle formation (Fig. 11a), while ALWC is very low (2.4 μg m$^{-3}$). This suggests that heterogeneous reaction might be involved into the new particle formation process under such NH$_3$-rich environments."

[Figure]

**Figure 11.** Evolution of meteorological parameters, secondary particulate matter (SPM), and size distribution during the two types of episodes (Ep2 and Ep5).

Line 380: Delete "e".

Deleted.

Line 393-395: What is the "different roles"? Please specify.

The different roles here refer to the potential influence of atmospheric processing (e.g., aqueous vs. photochemical processes) on SOA formation.

Line 398-399: What kind of SOA enhancement? I only saw an enhanced SOA at lower LWC concentration (<50 ug/m$^3$) in Fig 8a.

The enhancement of SOA under high RH conditions might be more-oxidized oxygenated organic aerosol, which could also be supported by the mass spectrum of OA with higher $f_{44}$ under high RH levels (fog events) rather than that under photochemical process dominated conditions (Please see Fig. S17 in the supplementary).

Line 419-423: The authors stated cooking emission included in the POA factor and POA showed clearly noon and night peaks for cooking emissions. Thus, the POA/CO ratios should not be dominated by urban traffic emissions. In contrast, I think the POA vs CO regression ratio reported here should be larger than POA vs CO ratio from urban traffic emissions, based on Hayes et al. 2013. What is the regression method for the POA/CO

calculation. Could the authors show the scatter plots. Orthogonal distance regression should be used. Is the intercept fitted to be zero or not?

We thank the reviewer's comments. Considering all reviewer comments and suggestions for reorganization of the manuscript, we have removed the original Figure about the relationship between OA and CO and related discussion. In the revised manuscript, we performed the linear regression of POA vs $\Delta CO$ (= CO – 0.3 mg m$^{-3}$) using orthogonal fit. By forcing the intercept to be zero, the slopes of PM$_1$ POA / $\Delta CO$ and PM$_{2.5}$ POA / $\Delta CO$ are 12.0 and 15.2, respectively, which are both higher than the typical value (6.4 µg m$^{-3}$/ppm, which is approximately 7.3 µg m$^{-3}$/mg m$^{-3}$) from traffic emissions (Hayes et al., 2013). These results support the influences of other primary emissions, e.g., cooking.

[Figure]

**Figure R5.** The relationship between POA and $\Delta CO$ in PM$_1$ and PM$_{2.5}$, respectively.

**References:**

Bae, M.-S., Demerjian, K. L., and Schwab, J. J.: Seasonal estimation of organic mass to organic carbon in PM$_{2.5}$ at rural and urban locations in New York state, Atmos. Environ., 2006, 40 (39): 7467-7479.

Cheng, Y., Zheng, G., Wei, C., Mu, Q., Zheng, B., Wang, Z., Gao, M., Zhang, Q., He, K., Carmichael, G., Pöschl, U., and Su, H.: Reactive nitrogen chemistry in aerosol water as a source of sulfate during haze events in China, Sci. Adv., 2, 10.1126/sciadv.1601530, 2016.

Hayes, P. L., Ortega, A. M., Cubison, M. J., Froyd, K. D., Zhao, Y., Cliff, S. S., Hu, W. W., Toohey, D. W., Flynn, J. H., Lefer, B. L., Grossberg, N., Alvarez, S., Rappenglück, B., Taylor, J. W., Allan, J. D., Holloway, J. S., Gilman, J. B., Kuster, W. C., de Gouw, J. A., Massoli, P., Zhang, X., Liu, J., Weber, R. J., Corrigan, A. L., Russell, L. M., Isaacman, G., Worton, D. R., Kreisberg, N. M., Goldstein, A. H., Thalman, R., Waxman, E. M., Volkamer, R., Lin, Y. H., Surratt, J. D., Kleindienst, T. E., Offenberg, J. H., Dusanter, S., Griffith, S., Stevens, P. S., Brioude, J., Angevine, W. M., and Jimenez, J. L.: Organic aerosol composition and sources in Pasadena, California, during the 2010 CalNex

campaign, Journal of Geophysical Research: Atmospheres, 118, 9233-9257, 10.1002/jgrd.50530, 2013.

Guo, H., Liu, J., Froyd, K. D., Roberts, J. M., Veres, P. R., Hayes, P. L., Jimenez, J. L., Nenes, A., and Weber, R. J.: Fine particle pH and gas–particle phase partitioning of inorganic species in Pasadena, California, during the 2010 CalNex campaign, Atmos. Chem. Phys., 17, 5703-5719, 10.5194/acp-17-5703-2017, 2017.

Li, Y. J., Sun, Y., Zhang, Q., Li, X., Li, M., Zhou, Z., and Chan, C. K.: Real-time chemical characterization of atmospheric particulate matter in China: A review, Atmos Environ, 158, 270- 304, https://doi.org/10.1016/j.atmosenv.2017.02.027, 2017.

Liu, M., Song, Y., Zhou, T., Xu, Z., Yan, C., Zheng, M., Wu, Z., Hu, M., Wu, Y., and Zhu, T.: Fine particle pH during severe haze episodes in northern China, Geophys Res Lett, n/a-n/a, 10.1002/2017gl073210, 2017.

Panteliadis, P., Hafkenscheid, T., Cary, B., Diapouli, E., Fischer, A., Favez, O., Quincey, P., Viana, M., Hitzenberger, R., Vecchi, R., Saraga, D., Sciare, J., Jaffrezo, J. L., John, A., Schwarz, J., Giannoni, M., Novak, J., Karanasiou, A., Fermo, P., and Maenhaut, W.: ECOC comparison exercise with identical thermal protocols after temperature offset correction – instrument diagnostics by in-depth evaluation of operational parameters, Atmos. Meas. Tech., 8, 779–792, doi:10.5194/amt-8-779-2015, 2015.

Petit, J. E., Favez, O., Albinet, A., and Canonaco, F.: A user-friendly tool for comprehensive evaluation of the geographical origins of atmospheric pollution: Wind and trajectory analyses, Environ. Modell. Softw., 88, 183-187, http://dx.doi.org/10.1016/j.envsoft.2016.11.022, 2017.

Sun, Y. L., Zhang, Q., Schwab, J. J., Demerjian, K. L., Chen, W. N., Bae, M. S., Hung, H. M., Hogrefe, O., Frank, B., Rattigan, O. V., and Lin, Y. C.: Characterization of the sources and processes of organic and inorganic aerosols in New York city with a high-resolution time-of-flight aerosol mass apectrometer, Atmos. Chem. Phys., 11, 1581-1602, 10.5194/acp-11-1581-2011, 2011.

Wang, G., Zhang, R., Gomez, M. E., Yang, L., Levy Zamora, M., Hu, M., Lin, Y., Peng, J., Guo, S., Meng, J., Li, J., Cheng, C., Hu, T., Ren, Y., Wang, Y., Gao, J., Cao, J., An, Z., Zhou, W., Li, G., Wang, J., Tian, P., Marrero-Ortiz, W., Secrest, J., Du, Z., Zheng, J., Shang, D., Zeng, L., Shao, M., Wang, W., Huang, Y., Wang, Y., Zhu, Y., Li, Y., Hu, J., Pan, B., Cai, L., Cheng, Y., Ji, Y., Zhang, F., Rosenfeld, D., Liss, P. S., Duce, R. A., Kolb, C. E., and Molina, M. J.: Persistent sulfate formation from London Fog to Chinese haze, Proc. Natl. Acad. Sci. U.S.A., 10.1073/pnas.1616540113, 2016.

Wang, Y. H., Liu, Z. R., Zhang, J. K., Hu, B., Ji, D. S., Yu, Y. C., and Wang, Y. S.: Aerosol physicochemical properties and implications for visibility during an intense haze episode during winter in Beijing, Atmos. Chem. Phys., 15, 3205-3215, 10.5194/acp-15-3205-2015, 2015.

Xu, W., Croteau, P., Williams, L., Canagaratna, M., Onasch, T., Cross, E., Zhang, X., Robinson, W., Worsnop, D., and Jayne, J.: Laboratory characterization of an aerosol chemical speciation monitor with PM2.5 measurement capability, Aerosol Sci. Technol., 51, 69-83, 10.1080/02786826.2016.1241859, 2017a.

Zhang, Q., Canagaratna, M. R., Jayne, J. T., Worsnop, D. R., and Jimenez, J.-L.: Time- and size-resolved chemical composition of submicron particles in Pittsburgh: Implications for aerosol sources and processes, J. Geophys. Res. Atmos., 110, 10.1029/2004JD004649, 2005.

Zhang, Y., Tang, L., Yu, H., Wang, Z., Sun, Y., Qin, W., Chen, W., Chen, C., Ding, A., Wu, J., Ge, S., Chen, C., and Zhou, H.-C.: Chemical composition, sources and evolution processes of aerosol at an urban site in Yangtze River Delta, China during wintertime, Atmos. Environ., 123, Part B, 339-349, http://dx.doi.org/10.1016/j.atmosenv.2015.08.017, 2015.

---

## Editor Decision (ED1)

The authors have reasonably addressed the comments of the two anonymous referees and they have modified their manuscript accordingly. However, numerous alterations are needed for the Main text and Supporting information before the manuscript can be published in ACP.

Main text:
Line 39: Replace "vaporizer of" by "vaporizer of the".
Line 45: Replace "also SOA" by "also the SOA".
Line 81: Replace "be the" by "be that the".
Line 83: Replace "lens, previous" by "lens, the previous".
Line 89: Replace "Therefore, the" by "Therefore,".
Line 95: Replace "and also" by "and are also".
Line 99: Replace "using EC-tracer" by "using the EC-tracer".
Line 111: Acronyms and abbreviations, here "SV" should be defined (written full-out) when first used. Thus, replace "of SV" by "of a standard vaporizer (SV)".
Line 115: Replace "of a standard vaporizer (SV)" by "of a SV".
Line 119: Replace "compositions" by "composition".
Line 122: Replace "measurements" by "measurement" and replace "compositions" by "composition".
Line 123: Replace "those measured" by "measurements".
Line 139: Replace "the Q-ACSM" by "the $PM_1$-Q-ACSM".
Line 151: Replace "the $PM_{2.5}$ Q-ACSM" by "the $PM_{2.5}$-Q-ACSM".
Line 154: Replace "of particle" by "of the particle".
Line 159: Replace "the size-selected" by "size-selected" and replace "particle" by "particles".
Line 160: Replace "respectively" by "respectively,".
Line 166: Replace "quantifications" by "quantification".
Line 172: Replace "into liquid" by "into the liquid".
Line 178: Replace "Particle" by "The particle".
Line 186: Replace "include the" by "included the".
Line 195: Replace "with PMF2.exe" by "with the PMF2.exe".
Line 198: Replace "in the both" by "in both" and replace "considering that" by "considering".
Line 199: Replace "(2) small" by "(2) a small".
Line 200: Replace "total organic signals" by "the total organic signal".
Line 204: Replace "supporting" by "the supporting".
Line 206: Replace "However higher" by "However, a higher" and replace "and multilinear" by "and the multilinear".
Line 211: Replace "was predicted" by "were predicted".
Line 225: Replace "except the" by "except for the".
Line 227: Replace "impacts on" by "impacts on the".
Line 228: Replace "fine" by "the fine".
Line 231: Replace "is due to the" by "is the".
Line 235: Replace "by Q-ACSMs" by "by the Q-ACSMs".
Line 242: Replace "between Q-ACSM and TDMPS" by "between the Q-ACSM and the TDMPS".
Line 250: Replace "including" by "including the".
Line 257: Replace "requires to" by "require to".
Line 261: Replace "that Q-ACSM" by "which the Q-ACSM".

Line 265: Replace "chloride" by "the chloride" and replace "explain by" by "explained by".
Line 267: Replace "concentrations also show" by "concentration also shows".
Line 268: Replace "between nitrate" by "between the nitrate".
Line 276: Replace "between" by "between the" and replace "that is" by "that is the".
Line 278: Replace "Fig.3b" by "Fig. 3b".
Line 279: Replace "previous studies generally reported" by "the previous studies generally reported a".
Line 281: Replace "in few" by "in a few".
Line 282: Replace "(with" by "(with a".
Line 290: Replace "up non-OA" by "up the non-OA".
Line 291: Replace "in future" by "in a future".
Line 298: Replace "those by" by "those measured by".
Line 303: Replace "in US" by "in the US".
Line 306: Replace "their large contributions" by "its large contribution".
Line 308: Replace "of aerosol species from" by "of the aerosol species from the".
Line 309: Replace "decrease as the" by "decrease with the".
Line 327: Replace "of PM$_{2.5}$ mass" by "of the PM$_{2.5}$ mass".
Line 342: Replace "entire periods" by "entire period".
Line 343: Replace "loadings during" by "loading during".
Line 345: Replace "contributions to" by "contribution to" and replace "that in" by "that to".
Line 347: Replace "those in" by "that in".
Line 350: Replace "particle in" by "particles in".
Line 369: Replace "by typical" by "by the typical".
Line 372: Replace "shows smaller" by "show smaller".
Line 380: Replace "are dominated" by "is dominated".
Line 382: Replace "be due to the" by "be the".
Line 383: Replace "produce non-OA" by "produce a non-OA".
Line 385: Replace "vaporizer in" by "vaporizer in the".
Line 386: Replace "release" by "release a".
Line 388: Replace "profiles in" by "profiles in the".
Line 389: Replace "also be the potential" by "also have a potential".
Line 392: Replace "on RH" by "on the RH".
Line 397: Replace "the later" by "the latter".
Line 409: Replace "were likely" by "was likely".
Line 411: Replace "increase as the increases" by "increase with the increase".
Line 412: Replace "increases of RH levels" by "RH levels increase".
Line 414: Replace "period with" by "a period with".
Line 423: Replace "variations of" by "variation of".
Line 428: Replace "rest time of" by "rest of".
Lines 431-432: This sentence needs to be rephrased; there is no main verb in it.
Line 436: Replace "reaction might" by "reactions might".
Line 444: Replace "concentrations of" by "concentration of".
Line 445: Replace "show dramatic" by "showed dramatic".
Line 450: Replace "play a" by "plays a".
Line 451: Replace "and it" by "and that it".
Line 458: Replace "with that during" by "with those during".

Line 467: Replace "concentrations of" by "concentration of" and replace "were measured" by "was measured".
Line 476: Replace "had insignificant" by "had an insignificant".
Line 477: Replace "PMF analysis" by "The PMF analysis".
Line 479: Replace "enhanced thermo" by "enhanced thermal".
Line 481: Replace "presented in" by "were present in" and replace "High" by "A high".
Line 482: Replace "and low ALWC was observed during new" by "and a low ALWC was observed during the new".
Line 487: Replace "remained relative" by "remained relatively".
Line 492: Replace "characterization of" by "characterization of the" and replace "compositions to" by "composition to".
Line 495: Replace "by Natural" by "by the Natural".
Line 498: Replace "supports on" by "support in".
Line 499: Replace "the PhD" by "a PhD".
Line 580: Replace "C., and Nenes" by "C. and Nenes".
Lines 810-824: "Zhang, J. K. et al., 2014" should come before "Zhang, Q. et al., 2007".
Lines 832-840: "Zhang, Y. et al., 2017" should come before "Zhang, Y. J. et al., 2015c".
Line 845: Replace "from TDMPS" by "from the TDMPS".
Line 857: Replace "dash line" by "dashed line".
Line 858: Replace "sum mass" by "summed mass".
Line 859: Replace "concentration of" by "concentration of the".
Line 860: Replace "that differences observed between" by "that the difference observed between the".
Line 866: Replace "species from" by "species from the".
Line 867: Replace "to standard" by "to the standard".
Line 868: Replace "solid squares" by "filled circles".
Line 876: Replace "in the (f) are coursed" by "in (f) are caused".
Line 882: Replace "(b) of" by "(b) of the".
Line 883: Replace "during different" by "during the different".
Line 888: Replace "inputs, for" by "inputs, for the".
Line 890: Replace ", and" by ", and the".
Line 901: Replace "SOA and" by "the SOA for NR-PM$_1$ and NR-PM$_{2.5}$ and".
Lines 901-902: If I understand Fig. 9c correctly, then "respectively. Note that more oxidized OOA (MO-OOA) was" should be replaced by "and between the more oxidized OOA (MO-OOA) and O$_x$ . Note that MO-OOA was".
Line 903: This sentence is unclear to me. Which calculation is meant here and in which way were the wet scavenging particles removed? Clarification and rewording are needed.
Line 908: The unit for the "wind speed" should be specified; I presume that it is "m s$^{-1}$".
Line 912: Replace "and size" by "and the size".
Line 913: Replace "during the two" by "during two".

Supporting information:
Page 2, captions of Figures S1 and S2: Replace "profiles of" by "profiles for the".
Page 3, caption of Figure S3, line 1: Replace "results for" by "results for the".
Page 3, caption of Figure S3, line 2: Replace "of number" by "of the number".

Page 3, caption of Figure S3, line 3: Replace "the box" by "box" and replace "of scaled" by "of the scaled".
Page 3, caption of Figure S3, line 4: Replace "a comparison" by "comparison".
Page 4, caption of Figure S4, line 1: Replace "results for" by "results for the".
Page 4, caption of Figure S4, line 2: Replace "of number" by "of the number".
Page 4, caption of Figure S4, line 3: Replace "the box" by "box" and replace "of scaled" by "of the scaled".
Page 4, caption of Figure S4, line 4: Replace "a comparison" by "comparison".
Page 5, caption of Figure S5, line 1: I do not understand why the word "Correction" is used here. Should it not be "Comparison" instead of "Correction"?
Page 5, caption of Figure S6, line 2: Replace "respectively." by "respectively. The".
Page 5, caption of Figure S6, line 3: Replace "system and" by "system and the".
Page 6, caption of Figure S8, line 1: I do not understand why "and/or" is used here. Should it not just be "and" instead?
Page 7, caption of Figure S9, line 1: Replace "particle" by "particles".
Page 7, caption of Figure S9, line 2: Replace "values of" by "value being".
Page 7, caption of Figure S10, line 1: Replace "between" by "between the" and replace "and" by "and the".
Page 7, caption of Figure S10, line 3: Replace "species of" by "species of the".
Page 7, caption of Figure S10, line 4: Replace "mass reports" by "mass accounts for".
Page 8, caption of Figure S11, line 1: Replace "between measured" by "between the measured".
Page 8, caption of Figure S11, line 2: Replace "in plots" by "in the plots".
Page 8, caption of Figure S11, line 3: Replace "predicted" by "the predicted".
Page 8, caption of Figure S12, line 3: Replace "which chemical" by "which the chemical".
Page 9, caption of Figure S13, line 1: Replace "between" by "between the".
Page 9, caption of Figure S13, line 2: Replace "with estimated" by "and the estimated" and replace "banlance" by "balance".
Page 9, caption of Figure S14, line 1: Replace "(a)" by "(a) the".
Page 9, caption of Figure S14, line 3: Replace "non-refractory" by "the non-refractory".
Page 10, caption of Figure S15, line 1: I do not understand why the word "correction" is used here. Should it not be "correlation" (or "comparison")" instead of "correction"? It is also unclear what the unit is in the abscissa and ordinate of Figs. (a') and (b').
Page 10, caption of Figure S15, line 2: Replace "from" by "from the".
Page 10, caption of Figure S16, line 1: Replace "of fine" by "of the fine".
Page 11, caption of Figure S17, line 1: Replace "OA for" by "OA for the".

---

## Author Response (AR3)

Dear Prof. Maenhaut,

We thank you very much again for your careful corrections. We have revised the manuscript and supplementary according to all your suggestions. Please find the track changes for the revised manuscript and supplementary in next pages.

Thank you very much for your time and we look forward to hearing back from you.

Sincerely yours,

Lili Tang & Yele Sun

[revised manuscript text omitted]

**Figure S10.** Correlations between the $PM_1$-ACSM, $PM_{2.5}$-ACSM, $PM_1$-BAM, $PM_{2.5}$-BAM and the volume-dependent mass (TDMPS and APS) with the particle density being calculated from the chemical species of the $PM_1$-ACSM and $PM_{2.5}$-ACSM, respectively. On average, the $PM_1$ and $PM_{2.5}$ Q-ACSM total dry mass accounts for respectively 89 % and 93 % of the $PM_1$ and $PM_{2.5}$ volume-dependent mass concentrations. As reported by Xu et al. (2017a), the $PM_{2.5}$ lens system showed a significant particle loss at below around 200 nm, with a lower transmission efficiency of 45 % on average. Considering this, we estimated that the lost of small particles at size ~13 – 201 nm might account for around 3 % of the total volume-dependent $PM_{2.5}$ mass (Fig. S10d).

[Figure]

**Figure S11.** Relationship between the measured NH4 and predicted NH4 for both the PM2.5 and PM1 ACSMs, respectively. The points in the plots are colored by the ratio of [SO4] / [SO4/NO3]. Note that the predicted NH4 is estimated by $18 \times (2\times[SO_4/96] + [NO_3/62] + [Cl/35.5])$.

[Figure]

**Figure S12.** Ion balance of the water-soluble ions measured by the PM2.5 MARGA. Note that: anion equivalents = $[NH_4^+/18] + [Na^+/23] + [K^+/39] + [Mg^{2+}/12] + [Ca^{2+}/20]$, and cation equivalents = $[SO_4^{2-}/48] + [NO_3^-/62] + [Cl^-/35.5]$, in which the chemical ions are in the unit of $\mu g\ m^{-3}$.

[Figure]

**Figure S13.** Relationship between the measured nitrate and chloride difference values (i.e., PM$_{2.5}$-Marga – PM$_{2.5}$-ACSM)  and the estimated maximum chloride by mass balance from Na$^{+}$, Ca$^{2+}$, K$^{+}$, and Mg$^{2+}$.

[Figure]

**Figure S14.** Relationships between (a) the PM$_1$ (measured by Met one BAM1020) and total PM$_{2.5}$ (measured by TEOM-FDMS and Met one BAM1020 respectively) mass loadings; and (b) the non-refractory NR-PM$_1$ (measured by the PM$_1$ ACSM) and PM$_{2.5}$ (NR-PM$_{2.5}$ measured by the PM$_{2.5}$-ACSM) for the entire study.

[Figure]

**Figure S15.** Time series (a-b) and correlation (a'-b') of the mass concentration *m/z* 60 and *m/z* 73 from the PM$_{2.5}$-ACSM and PM$_1$-ACSM, respectively.

[Figure]

**Figure S16.** Sized-segregated diurnal variations of the fine aerosol species and organic components.

[Figure]

**Figure S17.** Averaged mass spectra (MS) of OA for the $PM_1$ and $PM_{2.5}$ ACSM during the new particle formation (NPF, Episode 2) and the fog event (Fog, Episode 5) periods, respectively.

**Table S1.** Thermal protocol used in this study within the Sunset Lab. Semi-Continuous OC/EC Analyzer

| Gas | Hold time (s) | Temperature (°C) |
|---|---|---|
| He | 10 | 1 |
| He | 95 | 600 |
| He | 95 | 840 |
| He | 30 | Oven off |
| He | 5 | 550 |
| He/O$_2$ | 10 | 550 |
| He/O$_2$ | 25 | 550 |
| He/O$_2$ | 45 | 650 |
| He/O$_2$ | 115 | 870 |